# Childhood brain tumors instruct cranial hematopoiesis and immunotolerance

Elizabeth Cooper [1] ✉, David A. Posner[2,3], Colin Y. C. Lee [2,3], Linda Hu[1], Sigourney Bonner[1], Jessica T. Taylor [1], Oscar Baldwin [1,2], Rocio Jimenez-Guerrero[1], Katherine E. Masih [1,4], Katherine Wickham Rahrmann[1], Jason Eigenbrood [1], Gina Ngo[1], Valar Nila Roamio Franklin[1], Clive S. D'Santos[1], Richard Mair [1,5], Thomas Santarius[5], Claudia Craven[5], Ibrahim Jalloh[5], Julia Moreno Vicente [1], Timotheus Y. F. Halim [1], Li Wang [6], Arnold R. Kreigstien [6], Brandon Wainwright[7], Fredrik J. Swartling [8], Javed Khan [4], Menna R. Clatworthy [2,3] & Richard J. Gilbertson [1,9] ✉

Recent research has challenged a long-held view of the brain as an immune-privileged organ, revealing active immunosurveillance with therapeutic relevance. Using a new genetically engineered mouse model of *ZFTA–RELA* ependymoma, a childhood brain tumor, we characterized an immune circuit between the tumor and antigen-presenting hematopoietic stem and progenitor cells (HSPCs) in the skull bone marrow. The presentation of antigens by HSPCs to CD4[+] T cells biased HSPC lineages toward myelopoiesis and polarized CD4[+] T cells to regulatory T cells, culminating in tumor immunotolerance. Remarkably, normalizing hematopoiesis with a single infusion of antibodies directed against cytokines enriched in the cerebrospinal fluid of mice bearing *ZFTA–RELA* ependymomas, choroid plexus carcinomas or group 3 medulloblastoma—all aggressive childhood brain tumors—disrupted this process and caused profound tumor regression. These findings demonstrate the existence of a skull bone marrow–tumor immunological interface and suggest that modulating the local supply of myeloid cells could represent a less toxic therapeutic strategy for aggressive childhood brain tumors.

Almost all childhood brain tumors, which are the leading cause of childhood cancer death, are initiated in the embryonic brain[1]. These tumors retain the transcriptomic, morphological and functional characteristics of developing neural tissues, including the propagation of cell lineages from perivascular niches[2–4]. As they are initiated in utero before the immune system is fully mature[5], it is possible that they are tolerated as 'self'. Understanding how the immune system interacts with childhood brain tumors may therefore improve the use of alternative treatments, including immunotherapies.

Immunotherapies have had limited success in the treatment of childhood brain tumors, but they could ultimately prove more effective and less toxic than conventional surgery, radiation and chemotherapy[6]. The failure of immunotherapies to elicit an immune response in childhood brain tumors suggests a local source of immunosuppression, a notion supported by evidence that immune circuits suppress autoreactive inflammatory responses in the brain and certain adult brain tumors[3,7–12]. Indeed, cerebrospinal fluid (CSF) flows directly to the skull bone marrow, feeding immune cells with brain-derived signals and

driving immunosurveillance[11,13,14]; however, the mechanisms by which these signals influence hematopoiesis have remained unclear. Here we provide evidence that this circuit operates in childhood brain tumors and promotes the tolerance of these aggressive cancers, thereby identifying a therapeutic vulnerability.

## HSPCs in *ZFTA−RELA* ependymoma

To understand how immunosurveillance might operate in childhood brain tumors, we developed a genetically engineered mouse model of supratentorial ependymoma (EP$^{ZFTA-RELA}$) in which a conditional allele of the *ZFTA−RELA* fusion gene[14] (*Nestin$^{Flx-STOP-FlxZFTA-RELA}$*) is recombined in embryonic day 9.5 radial glia by the *Nestin$^{CreERT2}$* allele (Supplementary Fig. 1a−e and Supplementary Table 1): we originally identified radial glia as the cell of origin of ependymomas[2,14]. All *Nestin$^{CreERT2}$;Nestin$^{Flx-STOP-FlxZFTA-RELA}$* (*Nestin$^{Cre-ZFTA-RELA}$*) mice developed EP$^{ZFTA-RELA}$ tumors with a median survival of 90 ± 9.5 days; these tumors contain neural-progenitor-like cells highly enriched for a previously established human *ZFTA−RELA* ependymoma gene signature[2,14] (Supplementary Fig. 1f−h).

Flow cytometric analysis of both *Nestin$^{Cre-ZFTA-RELA}$* and *Nestin$^{CreERT2-WT}$* (control) mouse brains at embryonic day 12.5 identified similar populations of CD45$^+$ cells, including lineage-negative/Sca-1$^+$/c-Kit$^+$ (LSK$^+$) hematopoietic stem and progenitor cells (HSPCs) that had previously been described in normal embryonic mouse brain and human glioblastomas[15,16] (Fig. 1a,b, Supplementary Fig. 2 and Supplementary Tables 2 and 3). Following birth, CD45$^+$ cell populations in the brains of *Nestin$^{Cre-ZFTA-RELA}$* and control mice diverged markedly. By postnatal day 5 (P5), control mouse brains from which circulating CD45$^+$ cells were excluded during flow cytometry by CD45 intravenous labeling[17] lacked HSPCs and certain other immune cell types. By contrast, most immune cell populations, including HSPCs and regulatory T (T$_{reg}$) cells, persisted and/or expanded in *Nestin$^{Cre-ZFTA-RELA}$* brains. This began before appreciable tumor development (histologically undetectable before P19; Supplementary Fig. 1c) and persisted as tumors formed (Fig. 1a−d). Single-cell RNA sequencing (scRNA-seq) of normal human fetal brain[18,19] and 14 different types of human childhood brain tumor[20−24], including *ZFTA−RELA* ependymoma and group 3 medulloblastoma, confirmed the presence of similar immune populations in these tissues (Fig. 1e,f and Supplementary Fig. 3a−f).

HSPCs in human fetal brain and childhood brain tumors expressed major histocompatibility complex class II (MHC-II) and regulatory genes including *CITTA*, *HLA-DPA1*, *HLA-PB1*, *HLA-DRB1*, *HLA-DRB5*, *HLA-DQA1*, *HLA-DQA2* and antigen-loading chaperone *CD74* (Fig. 1g,h). Similarly, >90% of HSPCs in EP$^{ZFTA-RELA}$ tumors, but not LSK$^-$ committed progenitors, expressed MHC-II at levels similar to those of professional antigen-presenting cells (APCs; Fig. 1i,j, Extended Data Fig. 1a and Supplementary Table 4). MHC-II expressing HSPCs were also observed in tumor parenchyma, skull bone marrow and dura mater resected from a child with human choroid plexus papilloma (Supplementary Fig. 4a). HSPCs with antigen presentation capacity have been described previously in the peripheral bone marrow, where they can eliminate premalignant hematopoietic stem cells[25]; however, whether HSPCs with antigen presentation capacity exist in solid tissues and malignancies, including the brain, remained unknown.

To better understand the characteristics of brain-tumor-resident MHC-II$^+$ HSPCs, we first tested their long-term self-renewal capacity. Lineage-depleted CD45.1$^+$MHC-II$^+$ but not CD45.1$^+$MHC-II$^-$ cells isolated by fluorescence-activated cell sorting (FACS) from EP$^{ZFTA-RELA}$ tumors reconstituted hematopoiesis for up to 16 weeks in nonmyeloablative busulfan-conditioned mice carrying the CD45.2 allele, confirming the stem cell credentials of brain-tumor-resident MHC-II$^+$ HSPCs (Extended Data Fig. 1b−e).

Given the anatomical proximity of EP$^{ZFTA-RELA}$ tumors to the skull bone marrow, we considered whether local hematopoiesis might contribute immune cells to the tumor niche. Skull marrow is increasingly recognized as a source of immune cells to the central nervous system (CNS) in nonmalignant brain pathologies[26,27], prompting us to examine its involvement here. In EP$^{ZFTA-RELA}$-bearing mice, 5-ethynyl-2′-deoxyuridine (EdU) pulse labeling revealed a marked increase in proliferating HSPCs as well as total CD45$^+$ cells in the skull but not tibial (peripheral) bone marrow of postnatal EP$^{ZFTA-RELA}$-bearing mice relative to controls (Supplementary Fig. 5a,b). Monocytes, macrophages and immature B cells showed similar skull-restricted expansion (Supplementary Fig. 5c,d). This response extended to two additional childhood brain tumor models, choroid plexus carcinoma and group 3 medulloblastoma[28,29], each of which displayed increased levels of EdU$^+$ HSPCs relative to controls (Supplementary Fig. 5e,f). Thus, the presence of EP$^{ZFTA-RELA}$ tumors appeared to selectively engage the skull bone marrow niche, activating local hematopoiesis.

To characterize transcriptional changes within hematopoietic lineages in the presence of an EP$^{ZFTA-RELA}$ tumor, we generated scRNA-seq profiles of extravascular CD45$^+$ cells isolated from the skull, dura, deep cervical lymph nodes, tumor/brain and peripheral bone marrow of EP$^{ZFTA-RELA}$-bearing mice (n = 52,140 cells) and compared these with those of controls (n = 72,935 cells; Supplementary Fig. 6a−i). scRNA-seq profiles of skull bone marrow HSPCs, macrophages, neutrophils, monocytes and T$_{reg}$ cells isolated from EP$^{ZFTA-RELA}$-bearing mice were enriched for chemotaxis, cytokine signaling and myelopoiesis gene programs relative to controls (Supplementary Fig. 6h,i and Supplementary Tables 5−20). These changes were absent from peripheral bone marrow and deep cervical lymph nodes, indicating spatially confined reprogramming of immune activity within CNS proximal niches. Thus, the presence of an EP$^{ZFTA-RELA}$ tumor appeared to activate HSPCs in the skull bone marrow, potentially biasing hematopoiesis toward myelopoiesis and promoting chemotaxis.

## Brain-tumor-derived CSF cues inform skull hematopoiesis

CSF-borne proteins have been shown to educate the skull bone marrow to regulate CNS immune responses[7,30−33]. Therefore, we asked whether EP$^{ZFTA-RELA}$-derived cues might similarly modulate local hematopoiesis through the CSF.

In keeping with this notion, two independent proteomic profiling approaches detected significantly higher levels of myelopoietic (for example, G-CSF, GM-CSF), chemotactic (MIP-1a/b) and T cell activation/polarization (IL-12/23, IL-4) cytokines in the CSF of EP$^{ZFTA-RELA}$-bearing mice relative to controls (Fig. 2a and Extended Data Fig. 2a). Integration of these CSF profiles with scRNA-seq profiles of skull bone marrow CD45$^+$ cells identified potential CSF cytokine cross-talk with receptors on HSPCs, monocytes, neutrophils and B cells that could direct leukocyte migration, adhesion, integrin signaling and phagocytosis (Extended Data Fig. 2b−e and Supplementary Tables 21−24).

To test more directly whether CSF-borne signals could be carried to skull bone marrow HSPCs in our mice, we injected an anti-c-Kit antibody into the intrathecal space and looked for labeling of HSPCs (Fig. 2b,c and Supplementary Fig. 7a−d). Within 2 h of intrathecal injection, 85% and 60% of skull bone marrow and dural HSPCs, respectively, were labeled with anti-c-Kit regardless of tumor presence (Fig. 2c). Using a separate approach, we also showed that fluorescence-tagged ovalbumin (OVA), injected intrathecally, labeled F4/80$^+$CD64$^+$ skull bone marrow and dural macrophages (Fig. 2d and Supplementary Fig. 7e). HSPCs and macrophages in the tibial bone marrow were unlabeled.

We next considered whether tumor-derived cues in the CSF would direct skull bone marrow hematopoiesis and drive the migration of skull progeny into the tumor. To test this, we disrupted CXCR4-dependent mobilization by delivering AMD3100 either into the skull bone marrow or intrathecally in EP$^{ZFTA-RELA}$-bearing mice, followed by flow cytometric profiling of the brain, dura and skull bone marrow (Fig. 2e and Supplementary Fig. 7f). AMD3100, but not the control, significantly increased populations of macrophages, neutrophils and HSPCs in

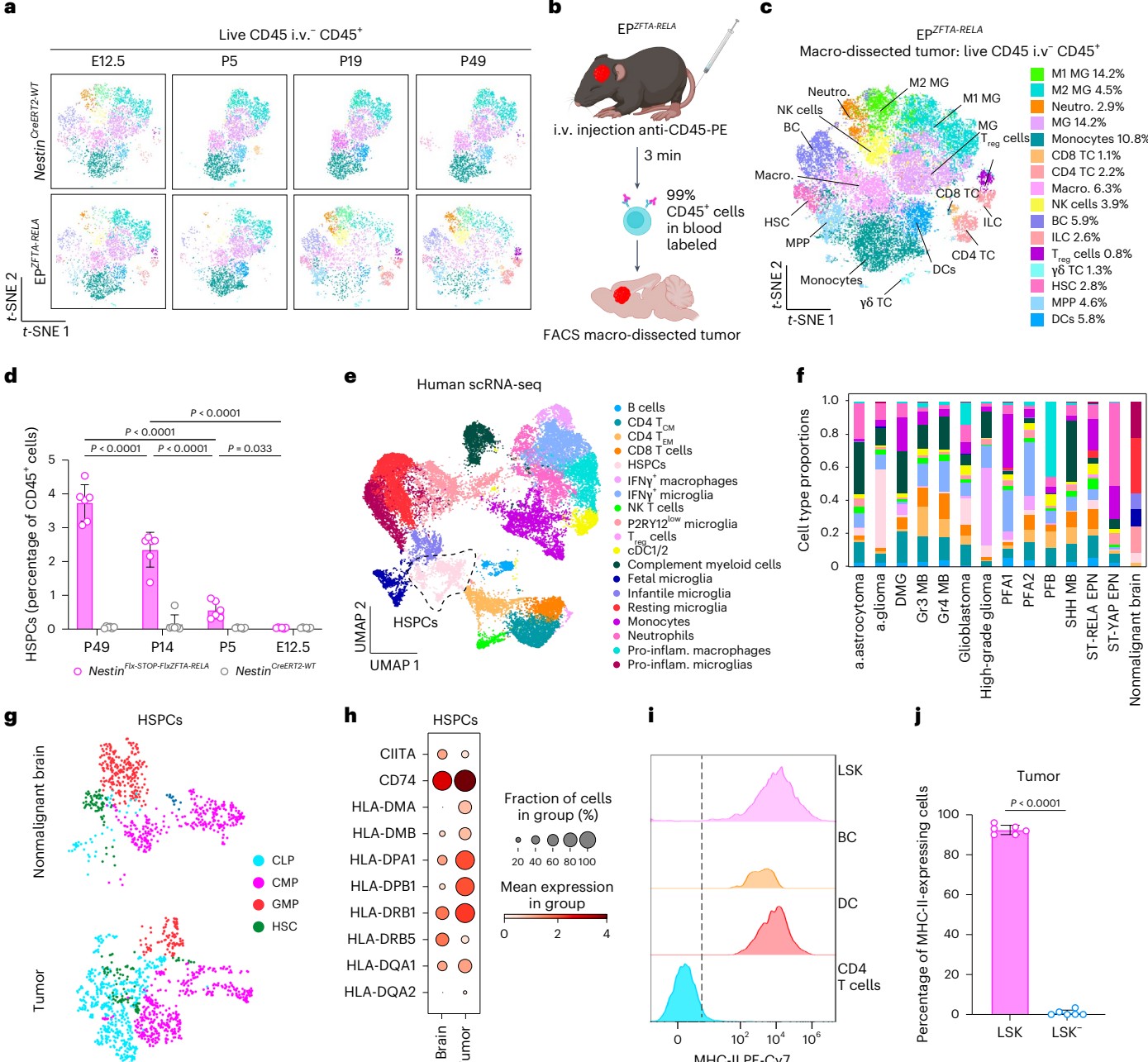

**Fig. 1 | Intratumoral MHC-II⁺ HSPCs in childhood brain tumors. a**, Representative *t*-distributed stochastic neighbor embedding (*t*-SNE) of flow cytometry data from developmental time points (embryonic day 12.5 (E12.5), P5, P19 and P49), with 3,000 events concatenated per sample, manually gated (key shown on the right in **c**). **b**, Schematic of the flow cytometry approach for characterizing the tumor microenvironment in a de novo *ZFTA–RELA* fusion ependymoma model. **c**, *t*-SNE analysis of 18,000 events from endpoint EP^*ZFTA-RELA* mice, showing proportions of CD45⁺ cells: M1 microglia, M2 microglia, neutrophils, microglia, monocytes, CD8 T cells, CD4 T cells, macrophages, natural killer cells, B cells, innate lymphoid cells, T_reg cells, γδ T cells, hematopoietic stem cells (HSCs), multipotent progenitor cells and dendritic cells. **d**, Quantification of LSK⁺ within the CD45⁺ population in EP^*ZFTA-RELA* and *Nestin^CreERT2* control mice (*n* = 5; mean ± s.e.m.; one-way analysis of variance (ANOVA) with Šídák's test). **e**, UMAP of human scRNA-seq data integrated from fetal brain tissue and tumor tissue from patients with childhood brain tumors, colored by cell type annotation: B cells; CD4 tissue central memory cells (T_CM); CD4 tissue effector memory cells (T_EM); CD8 T cells; HSPCs; IFNγ macrophages; IFNγ microglia; natural killer T cells; purinergic receptor P2Y12^low microglia; T_reg cells; conventional dendritic cell types 1 and 2; complement myeloid cells; fetal, infantile and resting microglia; monocytes; neutrophils; and proinflammatory macrophages and microglia. **f**, Quantification of proportions of cell types

across each disease group; anaplastic astrocytoma, anaplastic glioma, diffuse intrinsic pontine glioma, group 3 (Gr3) and group 4 (Gr4) medulloblastoma (MB), posterior fossa type A (PFA) ependymoma types 1 and 2, posterior fossa type B (PFB) ependymoma, sonic hedgehog (SHH) medulloblastoma, supratentorial-REL-associated protein (ST-RELA) ependymoma (EPN), ST-Yes1 associated transcriptional regulator (YAP) EPN. **g**, Tissue UMAP of HSPC clusters in malignant and nonmalignant brain tissue, colored by cell type annotation: common lymphoid progenitor (CLP), common myeloid progenitor (CMP), granulocyte–monocyte progenitor (GMP) and HSC. **h**, Dot plot of average and percentage expression of MHC-II antigen presentation machinery across HSPC cell clusters in malignant and nonmalignant brain. **i**, Representative histogram of MHC-II cell surface expression for LSK cells, B cells, dendritic cells and CD4 T cells. **j**, Quantification of proportions of MHC-II expression in intratumoral LSK⁺ relative to LSK⁻ cells (*n* = 6 per group, mean ± s.e.m., unpaired two-tailed Student's *t*-test). a.astrocytoma, anaplastic astrocytoma; a. glioma; anaplastic glioma; BC, B cells; cDC, conventional dendritic cells; DC, dendritic cells; DMG, diffuse intrinsic pontine glioma; i.v., intravenous; MG, microglia; MPP, multipotent progenitor cells; neutro., neutrophils; NK, natural killer; pro-inflam., proinflammatory; TC, T cells. Illustrations in **b** created with BioRender.com.

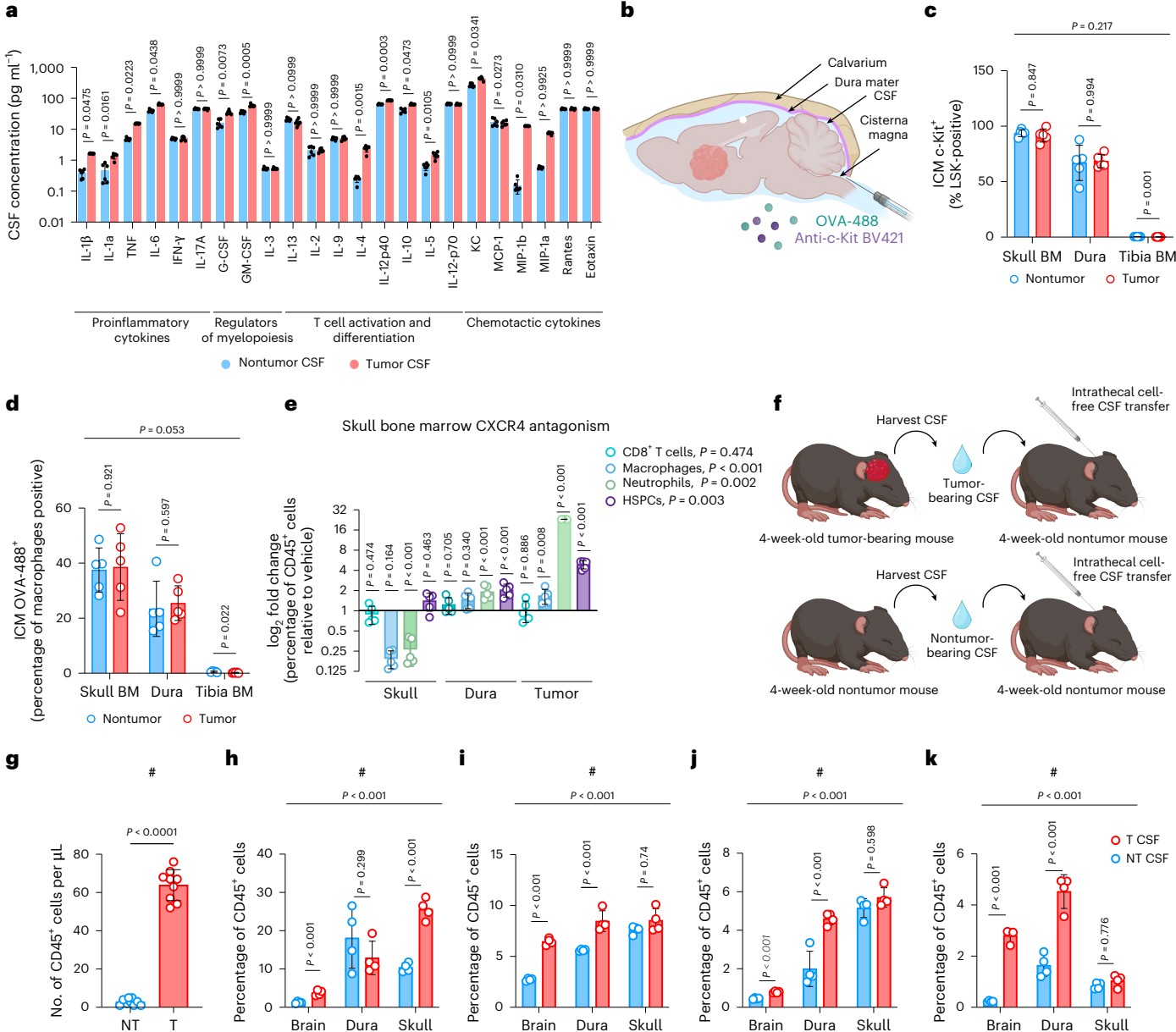

**Fig. 2 | Skull bone marrow cells access brain-derived solutes via the CSF and supply CNS tumors with HSPCs and myeloid cells. a,** Multiplexed measurement of cytokines and chemokines in CSF of EP$^{ZFTA-RELA}$-bearing and control mice using Luminex; $n = 6$ per group; data are mean ± s.e.m. $P$ values represent two-sided $t$-tests with Holm–Šídák multiplicity adjustment. **b,** Experimental design of intracisterna magna (ICM) injections of anti-c-Kit BV421 and OVA-488 into the CSF of tumor- and nontumor-bearing mice ($n = 5$ per group). **c,d,** Quantification of ICM-injected c-Kit$^+$Lin$^-$ Sca-1$^+$c-Kit$^+$ cells (**c**) and OVA$^+$ macrophages (**d**) in skull, tibia and dura ($n = 5$; mean ± s.e.m.; linear mixed-effects model with Wald $z$-tests and Bonferroni correction, two-sided). **e,** log$_2$ fold change of the

proportion of intratumoral macrophages, CD8 T cells, HSPCs and neutrophils following intracalvarial AMD3100 (2 mg kg$^{-1}$, 6 h) or aCSF treatment ($n = 5$ per group; mean ± s.e.m.; linear mixed-effects model with Wald $z$-tests and Bonferroni correction) relative to aCSF. **f,** Design for **g–k**: CSF transfer from tumor or nontumor mice ($n = 4$ per group). **g,** Quantification of CD45$^+$ cells in the CSF (mean ± s.e.m.; two-sided Student's $t$-test). **h–k,** Quantification of CD45$^+$ neutrophils (**h**), Ly6C$^-$ monocytes (**i**), HSPCs (**j**) and B cells (**k**) in dura, tibia and skull (mean ± s.e.m; linear mixed-effects model with Wald $z$-tests and Bonferroni correction, two-sided). BM, bone marrow; T, tumor; NT, nontumor. Illustrations in **b** and **f** created with BioRender.com.

EP$^{ZFTA-RELA}$ tumors and dura mater, whereas these populations decreased in the skull bone marrow. Again, no such changes were seen in the blood or peripheral bone marrow. Numbers of intratumoral and dural CD8$^+$ T cells were unaffected by AMD3100, consistent with a blood-trafficked origin or CXCR4-independent migratory pathway for these cells[34,35] (Fig. 2e).

To test whether signals present specifically in the CSF of EP$^{ZFTA-RELA}$-bearing mice could educate the skull bone marrow, we transferred cell-free CSF from EP$^{ZFTA-RELA}$-bearing or control mice into the intrathecal

space of naive age- and sex-matched C57BL/6 recipients and quantified changes in immune cell populations using flow cytometry (Fig. 2f). Six hours after CSF transfer, total numbers of CD45$^+$ cells in the CSF of mice receiving EP$^{ZFTA-RELA}$-donor CSF were significantly increased relative to those in controls, suggesting that the CSF is a likely route for tumor-driven skull-brain trafficking (Fig. 2g). This was associated with increased numbers of neutrophils, Ly6C$^-$ monocytes, HSPCs and B cells in the brains of mice receiving CSF from EP$^{ZFTA-RELA}$-bearing donors relative to controls (Fig. 2h–k).

Together, these data support the hypothesis that EP[ZFTA-RELA]-derived signals are carried in the CSF to the skull bone marrow, promoting mobilization of HSPCs and myeloid cells to the dura and tumor.

## Dural sinuses are hubs of antitumor immunosurveillance

During normal brain homeostasis and neuroinflammation, the dura is a key site of CNS immune surveillance[27,30,36,37], where CSF-derived antigens are drained via meningeal lymphatics to facilitate presentation to T cells in the dura mater and draining lymph nodes[13,30]. Therefore, we investigated whether the dura might be a site of immune activation in EP[ZFTA-RELA]-bearing mice. These mice exhibited striking lymphoid aggregates around the confluence of sinuses, bridging veins and the rostral–rhinal hub, regions of known CSF–lymphatic interface[36]. These aggregates contained dense infiltrates of CD4+ and CD8+ T cells, IBA1+ macrophages and MHC-II+ APCs, the populations of which were significantly expanded relative to those of control mice (Fig. 3a–h). Dural aggregates also showed elevated levels of IFNγ-expressing CD4+ T cells, T helper 1 ($T_H1$) cells and $T_{reg}$ cells, and increased IFNγ secretion was confirmed in fresh dural explants (Fig. 3i–k). Single-cell T cell receptor (TCR) analysis identified expanded CD4+ and CD8+ T cell clones shared between skull marrow and dura, suggesting coordinated antitumor responses (Extended Data Fig. 3a–i).

HSPCs (Sca-1+Kit+CD150+) were similarly enriched in the dura (Extended Data Fig. 4a,b). In contrast to lymphoid aggregates, which formed around venous sinuses, HSPCs were located adjacent to nonsinus blood vessels, suggesting that unlike skull HSPCs, dural HSPCs might monitor peripheral signals. Direct contact between APCs and CD3+ T cells was significantly more frequent in the dura of EP[ZFTA-RELA]-bearing mice than in that of control mice, and scRNA-seq profiles of CD45+ cells isolated from the dura of EP[ZFTA-RELA]-bearing mice were significantly more enriched for regulators of antigen receptor signaling and T cell differentiation pathways relative to those from control mice (Extended Data Fig. 4c,d and Supplementary Tables 7 and 8).

Thus, consistent with observations in skull bone marrow, the immune composition and landscape of the dura were markedly altered by the presence of an EP[ZFTA-RELA] tumor, supporting the notion that the dura serves as a site of tumor antigen engagement.

## Brain tumors drive $T_{reg}$ cell polarization in the skull bone marrow

Having established that MHC-II+ HSPCs in EP[ZFTA-RELA]-bearing mice were pluripotent, skull-derived and responsive to CSF cues, we next asked whether they possessed antigen-presenting capacity. We used the Y-Ae monoclonal antibody, which recognizes presentation of the exogenous $E\alpha_{52-68}$ peptide on I-A^b (ref. 38; Fig. 4a). EP[ZFTA-RELA]-bearing and control mice received intrathecal Eα-peptide, and peptide presentation in the skull bone marrow, dura and tumor was quantified by Y-Ae flow cytometry. As expected, CD11c+ dendritic cells but not CD4+ T cells in EP[ZFTA-RELA] tumors efficiently presented the Eα-peptide via MHC-II (Fig. 4b). Notably, HSPCs in the tumor and skull marrow also robustly presented the MHC-II-restricted peptide, mirroring the APC-like HSPCs described in peripheral marrow[25], as further demonstrated in a DQ-OVA assay (Fig. 4b–d). This antigen presentation was blocked in vitro by preincubation of HSPCs isolated from EP[ZFTA-RELA] tumors with anti-MHC-II antibody before treatment with Eα-peptide, confirming that these cells present exogenous peptides via MHC-II (Extended Data Fig. 5a–c).

A defining feature of APCs is their capacity to activate CD4+ T cells in an antigen-specific manner; therefore, we investigated whether HSPCs in EP[ZFTA-RELA]-bearing mice might present antigens to CD4+ T cells. To do this, we made use of CD4+ T cells isolated from OT-II mice that specifically recognize the OVA peptide ($OVA_{323-339}$) in the context of MHC-II[25] (Fig. 4e). Skull bone marrow HSPCs, dendritic cells and CD8+ T cells were isolated by FACS from EP[ZFTA-RELA]-bearing and control mice and incubated ex vivo with $OVA_{323-339}$. HSPCs and

dendritic cells, but not CD8+ T cells, robustly activated OT-II CD4+ T cells in vitro, regardless of tumor presence, at levels similar to those achieved with these cells isolated from tibial bone marrow (Fig. 4f and Extended Data Fig. 5d). T cell activation was correlated with increasing T cell/HSPC ratios and could be abrogated by preincubation of HSPCs with anti-MHC-II antibody, confirming the MHC-II dependency of this interaction (Extended Data Fig. 5e–g). Direct HSPC–CD4+ T cell interaction was required for HSPCs to activate T cells (Extended Data Fig. 5h). To ensure that skull antigen presentation by HSPCs was not restricted to OVA-OT-II interactions, we confirmed activation of CD4+ T cells in an MHC-II-dependent manner using FACS-isolated HSPCs inoculated with a different CNS antigen (myelin oligodendrocyte glycoprotein [MOG]$_{35-55}$; Extended Data Fig. 5i,j). Thus, similar to those previously identified in long bones[25], HSPCs with APC capacity exist in postnatal skull bone marrow; however, these cells appear to exist in the postnatal brain only in the presence of a brain tumor.

To determine how antigen presentation by HSPCs might affect tumor surveillance, we investigated whether this resulted in polarization of naive CD4+ T cells into proinflammatory or immunosuppressive T helper subsets. $OVA_{323-339}$ and $MOG_{35-55}$ HSPCs primed ex vivo, which we isolated from skull bone marrow and tumors, potently upregulated FOXP3 expression in OT-II and 2D2 T cells, indicating $T_{reg}$ cell polarization (Fig. 4g and Extended Data Fig. 5k–m). Furthermore, scRNA-seq profiles of CD45+ cells isolated from the skull and dura of EP[ZFTA-RELA]-bearing mice were significantly enriched for genes associated with $T_{reg}$ cell polarization relative to those from control mice (Supplementary Fig. 6h and Supplementary Tables 19 and 20). HSPCs exhibited low-to-moderate expression of conventional costimulatory molecules but markedly elevated levels of coinhibitory receptor PD-L1 (Supplementary Fig. 6i). Furthermore, comprehensive flow cytometry profiling of central and peripheral bone marrow sites in EP[ZFTA-RELA]-bearing and control mice revealed selective enrichment of FOXP3+CD4+ T cells in the skull bone marrow of EP[ZFTA-RELA]-bearing mice (Extended Data Fig. 5n,o). These data suggest that HSPCs derived from skull bone marrow function as APCs, but expression of coinhibitory molecules such as PD-L1 modulate T cell responses towards immunosuppression.

To test directly whether CD4+ T cells in the skull bone marrow were polarized toward $T_{reg}$ cells in response to brain-derived peptides in vivo, we used an adoptive transfer model in RAG2-knockout mice that lacked mature B and T cells (Extended Data Fig. 6). CD90.1+CD4+FOXP3− cells were adoptively transferred into RAG2-knockout mice, which were then injected intrathecally at days 14 and 17 posttransfusion with either artificial CSF (aCSF; vehicle) or 10 μg of $MOG_{35-55}$. Donor cells isolated 38 days postinfusion from the dura and skull of MOG-injected mice demonstrated a marked increase in CD4+ T cell polarization toward FOXP3+ $T_{reg}$ cells relative to those from aCSF-treated mice (Extended Data Fig. 6b–e). No such polarization was observed among cells harvested from the spleen, blood or brain. We were able to corroborate this effect in wild-type mice, demonstrating that CNS-derived peptides could also drive the polarization of conventional CD4+ T cells into $T_{reg}$ cells within the skull (Extended Data Fig. 6f–i).

Given the response of naive CD4+ T cells to CNS antigens in the skull bone marrow, we reasoned that CNS tumor neoantigens might favor the same immunosuppressive response. To test this, we identified a MHC-II (I-A/E) predicted neoantigen unique to the ZFTA–RELA fusion protein in our EP[ZFTA-RELA] mouse model and inoculated EP[ZFTA-RELA]-bearing and control mice with this peptide (Fus1$_{360-71}$), $MOG_{35-55}$ or aCSF intrathecally (Fig. 4h and Supplementary Table 25). Consistent with a classical response to a foreign antigen in control mice, Fus1$_{210-24}$ failed to promote $T_{reg}$ cell polarization in the skull bone marrow and favored a $T_H1$ response, as indicated by elevated levels of IFNγ+CD4+ T cells (Fig. 4i,j). Notably, mice harboring EP[ZFTA-RELA] tumors inoculated with Fus1$_{210-24}$ mirrored the response of mice inoculated with CNS self-peptide $MOG_{35-55}$, including $T_{reg}$ cell polarization and a lack of IFNγ+CD4+ T cell induction.

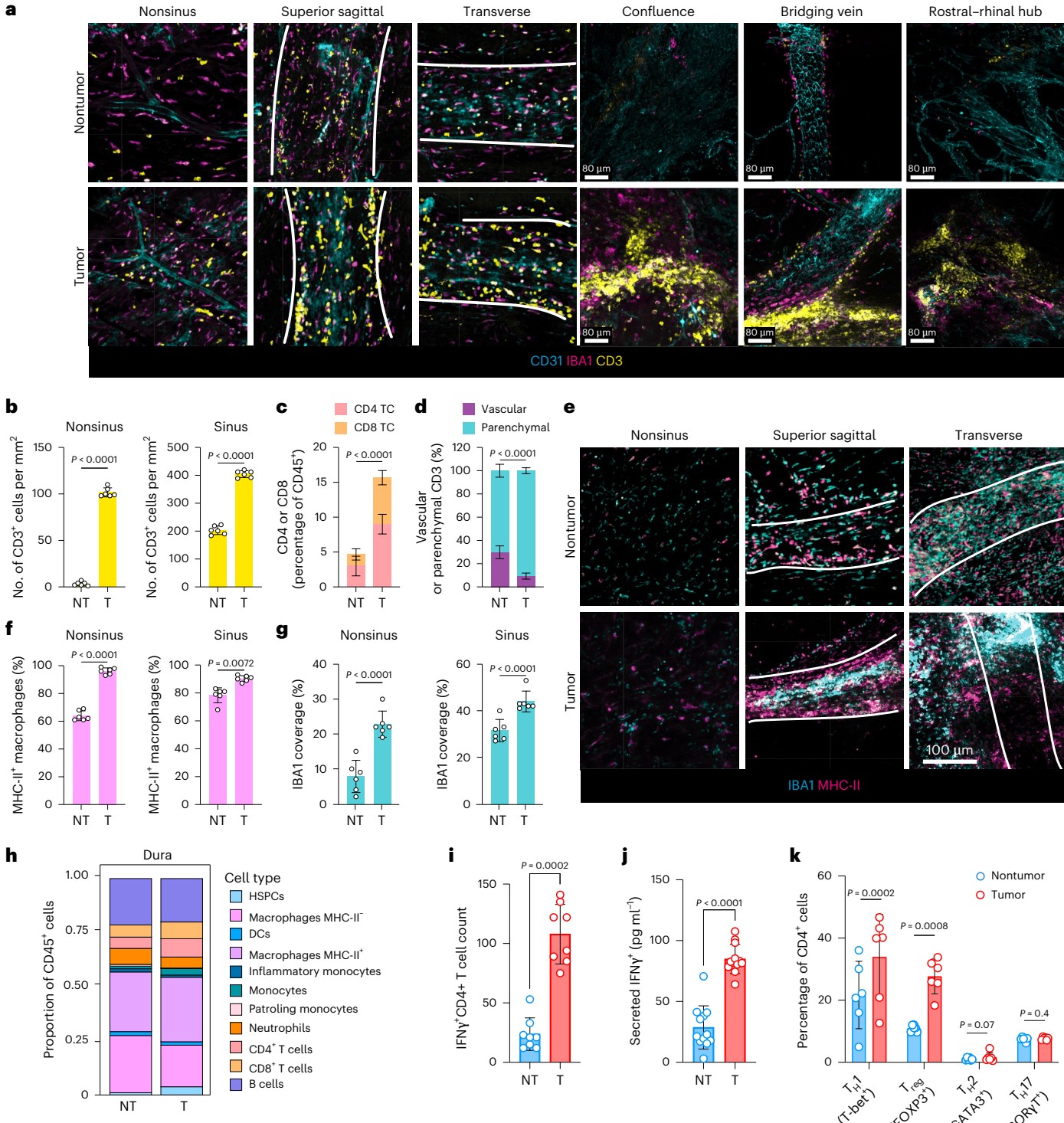

**Fig. 3 | Dural sinuses are regional hubs for meningeal antitumor immunosurveillance. a**, Immunohistochemistry of CD3⁺ T cells, IBA1⁺ macrophages and CD31⁺ endothelium at key regions of immunosurveillance in the dura mater in tumor-bearing and control mice. **b,c**, Quantification of numbers (**b**) and proportions (**c**) of CD3⁺, CD4⁺ and CD8⁺ T cells at dural sinus and nonsinus in EP*ZFTA-RELA*-bearing and control mice. **d**, Flow cytometry quantification of the proportion of vascular (CD45 i.v.⁺) or parenchymal (CD45 i.v.⁻) CD3⁺ T cells in EP*ZFTA-RELA*-bearing and control mice; *n* = 6. **e**, Immunohistochemistry of MHC-II⁺IBA1⁺ macrophages in sagittal sinus, transverse sinus and nonsinus regions of the dura mater. **f,g**, Quantification of proportions of IBA1⁺ cells that coexpressed MHC-II (**f**) and the percentage area coverage of IBA1 immunoreactivity (**g**) in

the nonsinus and perisinus regions in EP*ZFTA-RELA*-bearing mice (*n* = 6 per group; mean ± s.e.m.; unpaired two-tailed Student's *t*-test). **h**, Flow cytometry analysis and quantification of proportions of immune cell types in the dura (*n* indicates the average of 12 mice per group). **i**, Quantification of meningeal CD4⁺IFNγ⁺ T cells in tumor-bearing and control mice (*n* = 8 mice per group, 3 independent experiments). **j**, Quantification of IFNγ in culture supernatants following ex vivo stimulation of dural whole mounts (*n* = 10 mice per group, 2 independent experiments). **k**, Flow cytometry analysis and quantification of T cell phenotypes; T_H1, T_reg, T_H2 and T_H17 cells in the meningeal dura mater of EP*ZFTA-RELA*-bearing and control mice (*n* = 6; mean ± s.e.m.; one-way ANOVA with Šídák's test).

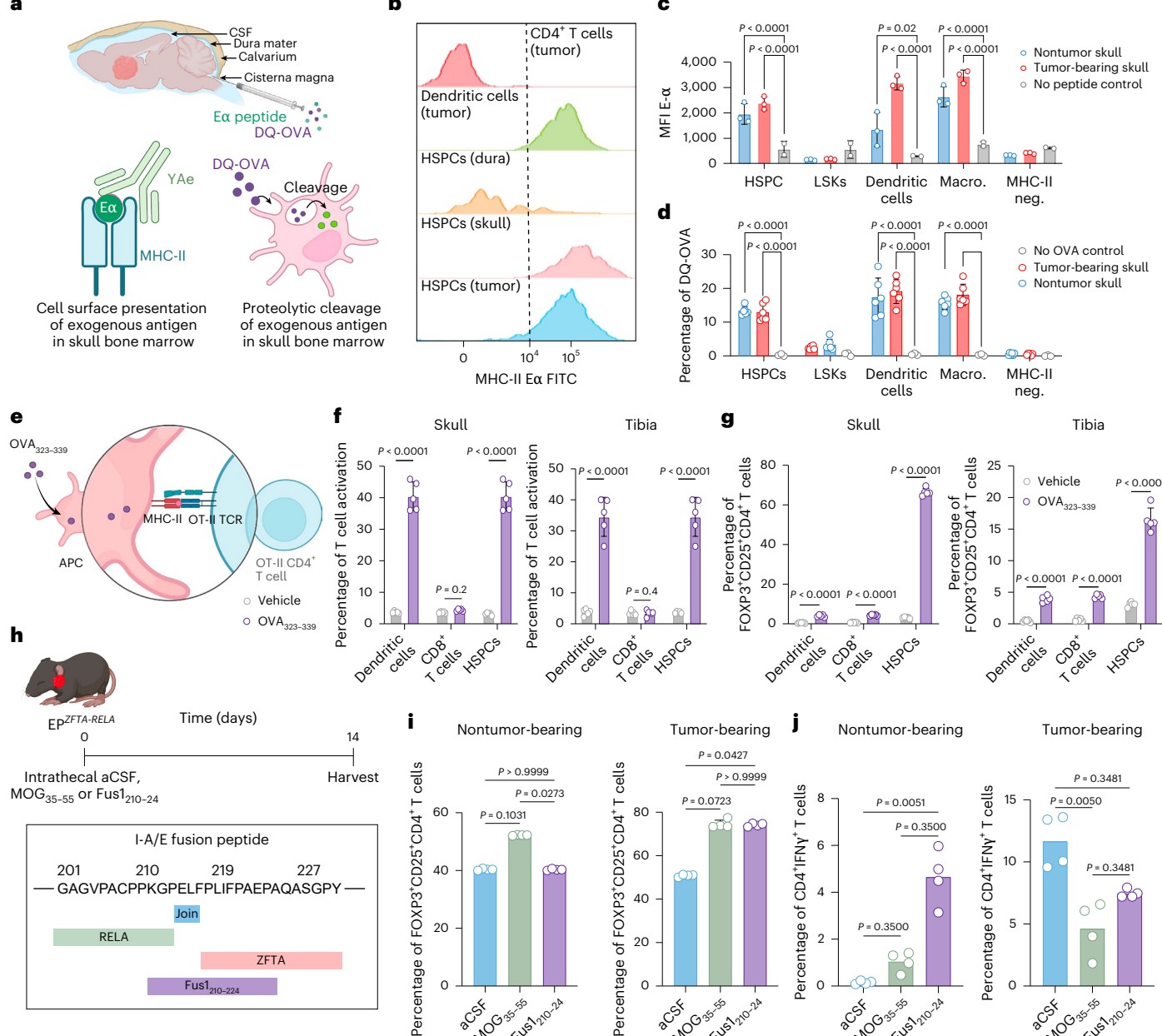

**Fig. 4 | Antigen presentation of CNS and tumor neoantigen drives T$_{reg}$ cell polarization in the skull bone marrow. a**, Experimental design for profiling of antigen uptake and processing via ICM injections of exogenous peptides. **b**, Flow cytometry histograms of MHC-II Eα immunoreactivity in FACS-isolated CD4$^+$ T cells, B cells, dendritic cells and HSPCs from tibia and skull bone marrow after Eα peptide immunization (500 µg ml$^{-1}$ i.v., 5 h). **c**, Quantification of MHC-Eα$^+$ cells by genotype with or without anti-MHC-II antibody ($n$ = 3, 10 mice per replicate; mean ± s.e.m.; one-way ANOVA with Šídák's test). **d**, Proportions of DQ-OVA$^+$ cells in tumor and nontumor mice following ICM injection of DQ-OVA for 2 h ($n$ = 5, 10 mice per replicate; mean ± s.e.m.; one-way ANOVA with Šídák's test). **e**, Experimental design for ex vivo peptide pulsing with vehicle or OVA$_{323-339}$.

**f**, T cell activation (CD44$^+$ cells) after coculture of skull- or tibia-derived dendritic cells, CD8$^+$ T cells and HSPCs with OT-II CD4$^+$ T cells ($n$ = 5, 10 pooled mice per replicate; mean ± s.e.m.; one-way ANOVA with Šídák's test). **g**, FOXP3 expression in vehicle or OVA-immunized dendritic cells, CD8$^+$ T cells and HSPCs after OT-II CD4$^+$ T cell coculture ($n$ = 5). **h**, Experimental design for intrathecal injection of aCSF, MOG$_{35-55}$ or Fus1$_{210-24}$. **i**, FOXP3 expression in skull CD4 T cells in immunized mice 14 days after injection with aCSF, MOG$_{35-55}$ or Fus1$_{210-24}$. **j**, Quantification of skull CD4$^+$IFNγ$^+$ T cells following intrathecal injection in tumor-bearing or control mice ($n$ = 4 per group, 2 independent experiments). MFI, mean fluorescence intensity. Illustrations in **a**, **e** and **h** created with BioRender.com.

Hence, in the context of EP$^{ZFTA-RELA}$, the fusion neoantigen is recognized as a self-antigen by the developing skull bone marrow. In further support of this, only 4.8% of EP$^{ZFTA-RELA}$ tumor cells engrafted in the brains of syngeneic adult C57BL/6 mice, compared to an engraftment rate of 98.5% in perinatal pups ($χ^2$ = 96.685, $P$ < 0.00001, and data not shown), strongly supporting the notion that local and early immunotolerance is important in tumorigenesis. Thus, skull HSPCs and other APCs take

up, process and present CSF antigens via MHC-II, activating and polarizing T cells to provide a local supply of immunosuppressive T$_{reg}$ cells.

## Ependymoma drives aberrant myelopoiesis in the skull bone marrow

We hypothesized that the presence of a brain tumor, in addition to subverting HSPC–T cell interactions, would reshape lineage commitment

of skull bone marrow HSPCs. As a first step to test this, we interrogated the regulatory mechanisms underpinning hematopoiesis using single-nuclear assay for transposase-accessible chromatin with sequencing (snATAC-seq) and transcriptomic analysis of CD34$^+$LSK$^+$ cells that we had FACS-isolated from the skull and peripheral bone marrow of EP$^{ZFTA-RELA}$-bearing and control mice. Uniform manifold approximation and projection (UMAP) of chromatin accessibility landscapes revealed discrete clustering of hematopoietic subpopulations, with skull HSPCs from EP$^{ZFTA-RELA}$-bearing mice displaying a marked shift toward myeloid fate commitment relative to controls (Fig. 5a,b and Supplementary Fig. 8a–f). Pseudotime analysis further supported a trajectory marked by reduced lymphoid output and dominance of myeloid programs (Fig. 5c,d). Motif enrichment analysis across pseudotime revealed a myeloid-specification program that included genes downstream of GM-CSFRα—for example, *Fos*, *Bach1/2* and *Smarcc1*—consistent with the elevated GM-CSF levels we had observed in the CSF (Figs. 2a and 5e,f and Extended Data Fig. 2a), whereas gene ontology analysis demonstrated concomitant suppression of lymphoid-associated program in skull relative to peripheral bone marrow (Fig. 5g,h). Together, these data suggest that tumor-derived signals in the CSF reprogram local HSPCs toward a myeloid-skewed state at the expense of lymphopoiesis.

To validate the myeloid bias observed in tumor-associated skull HSPCs, we employed two complementary approaches. First, colony-forming unit assays confirmed that HSPCs isolated from the skull bone marrow of EP$^{ZFTA-RELA}$-bearing mice produced significantly more granulocyte–macrophage colonies than those from controls, consistent with a myeloid bias fate (Supplementary Fig. 8g). Second, OVA-primed skull HSPCs that we cocultured with OT-II CD4$^+$ T cells ex vivo differentiated toward granulocyte–monocyte progenitor and myeloid-lineage cells (Fig. 5i). This cellular reprogramming was accompanied by increased production of IL-10, nitrite (NO$_2^-$) and reactive oxygen species (ROS), consistent with a myeloid-derived suppressor cell (MDSC)-like phenotype[39] (Fig. 5j–l).

Finally, we further tested the contribution of HSPC antigen presentation to tumor progression by generating an inducible MHC-II knockout mouse in which H2-Ab1$^{tm1Koni/J}$ mice were bred to SLC-CreER$^{T2}$ mice (Extended Data Fig. 7a). Tamoxifen induction at P0 and P1 achieved 60% knockdown of MHC-II expression on LSK$^+$ cells within the skull bone marrow, which was partially restored by P21 (Extended Data Fig. 7a,b). The growth of orthotopic EP$^{ZFTA-RELA}$ tumor allografts in these mice was significantly delayed relative to those in mice with intact MHC-II, resulting in prolonged survival (median survival: 67 days versus 37.5 days, $P = 0.0025$; Extended Data Fig. 7c–e).

Together, these findings indicate that EP$^{ZFTA-RELA}$ tumors reprogram local HSPC chromatin and transcriptional landscapes in favor of myeloid lineages, linking skull marrow hematopoiesis to tumor-induced immune remodeling. Sustained antigen presentation by skull HSPCs promotes myelopoiesis and immune suppression in the context of EP$^{ZFTA-RELA}$ tumors, indicating that this axis is a driver of local immunotolerance.

## Targeting CNS immunosurveillance in childhood brain tumors

Given the profound relationship between local immunotolerance and brain tumorigenesis, we considered whether this might represent a therapeutic vulnerability of childhood brain tumors. We reasoned that elevated GM-CSF in the CSF, which is likely to be produced by cells across the tumor, dura and skull, would participate in a circuit in which tumor-derived signals drive aberrant hematopoiesis in adjacent marrow (Supplementary Fig. 9a,b). Indeed, analysis of our transcriptomic data identified high-level expression of key myelopoiesis regulator *CSF2RA* (encoding GM-CSFRα) in the skull bone marrow (Supplementary Fig. 9c,d). Although not exclusive to HSPCs, this receptor is well positioned to convert local GM-CSF signals into a myelopoietic program[40,41]. Therefore, we randomized mice bearing EP$^{ZFTA-RELA}$ tumors (confirmed by bioluminescence exploiting the *Nestin*$^{Cre-ZFTA-RELA}$–IRES–luciferase allele; Supplementary Fig. 1) to receive a single intrathecal injection of anti-GM-CSF antibody (5 mg kg$^{-1}$), mavrilimumab (5 mg kg$^{-1}$; an anti-GM-CSFRα antibody) or control antibody (Fig. 6a). Remarkably, single-dose antibody blockade of either GM-CSF or its receptor induced near-complete regression of EP$^{ZFTA-RELA}$ tumors that was sustained for around 6 weeks and associated with a greater than threefold increase in survival time (Fig. 6b,c and Supplementary Table 28). GM-CSF targeted therapy also decreased CSF GM-CSF levels, skull HSPC proliferation and tumor-associated myeloid cells, whereas tumor CD8$^+$ T cell infiltration increased within 21 days (Extended Data Fig. 8a–h).

Given the evidence of aberrant skull marrow proliferation in multiple childhood brain tumor models (Supplementary Fig. 5e,f), we considered whether the GM-CSF signaling axis might represent a shared therapeutic vulnerability. Remarkably, mavrilimumab therapy in choroid plexus carcinoma (CPC) and group 3 medulloblastoma[28,29] models elicited profound tumor suppression, reduced CSF GM-CSF, decreased skull HSPC and tumor myeloid cells, and increased CD8$^+$ T cells (Extended Data Fig. 8i–t). These data position GM-CSF as a central driver of skull marrow myelopoiesis and tumor tolerance across diverse pediatric brain tumors and indicate the potential of skull-directed therapy for intracranial malignancy.

The rapid tumor regression and influx of CD8$^+$ T cells following GM-CSF blockade suggested that skull-derived myeloid suppressor cells constrain cytotoxic immunity. In line with this, immunohistochemistry and flow cytometry showed increased intratumoral CD8$^+$ T cells that persisted at relapse (Fig. 6e,f). Furthermore, TCR profiling revealed clonal expansion of CD8$^+$ and CD4$^+$ T cells, and gene ontology analysis of CD8$^+$ T cell populations revealed upregulation of type 1 interferon response, regulation of cell killing and T cell activation (Fig. 6g–j, Supplementary Fig. 10a–h and Supplementary Tables 29–34). CD8$^+$ T cells with high clonal expansion displayed elevated cytotoxicity (GZMK, GXMB), activation (NKG7, CCL5), inflammation and chemotaxis (CCR5, XCL1, S100A6) and exhaustion (LILRB4a, PDCD1; Extended Data Fig. 9) of their intratumoral immune populations—including monocytes, innate lymphoid cells and microglia—as well as upregulation of antigen presentation, myeloid activation and interferon-responsive pathways, consistent with broad microenvironmental reprogramming (Supplementary Tables 35–44). Consistent with an on-target effect, intratumoral HSPCs downregulated hematopoiesis, lymphocyte differentiation and immune system development programs following mavrilimumab treatment (Supplementary Tables 35 and 36). Together, these findings reveal that skull-marrow-directed GM-CSF blockade diminishes the supply of MDSCs and promotes clonal expansion and activation of tumor-infiltrating T cells.

Relapsed tumors showed increased expression of exhaustion markers (*Ctla4*, *Lag3*, *Pdcd1*, *Tigit*), suggesting that immune exhaustion contributes to therapeutic failure (Fig. 6k, Supplementary Fig. 10i and Supplementary Tables 29–46). We therefore tested whether anti-CTLA-4 could augment GM-CSF blockade. Intravenous mavrilimumab (10 mg kg$^{-1}$) alone markedly prolonged survival ($P < 0.0001$, $n = 11$), and combining it with anti-CTLA-4 (3 mg kg$^{-1}$) further improved efficacy ($P < 0.0223$, $n = 11$) (Fig. 6l and Supplementary Table 47), whereas anti-CTLA-4 alone conferred no benefit ($P < 0.4379$, $n = 11$).

Despite systemic administration, peripheral immune compartments remained unchanged (Extended Data Fig. 10). By contrast, mavrilimumab, alone or in combination, significantly reduced intratumoral monocytes, B cells, neutrophils and CD4$^+$ T cells (all $P < 0.001$), while increasing CD8$^+$ T cells in tumor parenchyma ($P < 0.001$) and dura ($P = 0.037$). Skull marrow macrophages, monocytes, neutrophils and T cell subsets were diminished, with compartmental interactions indicating redistribution of immune cells from skull marrow toward tumor and dura. These data show that normalizing skull marrow

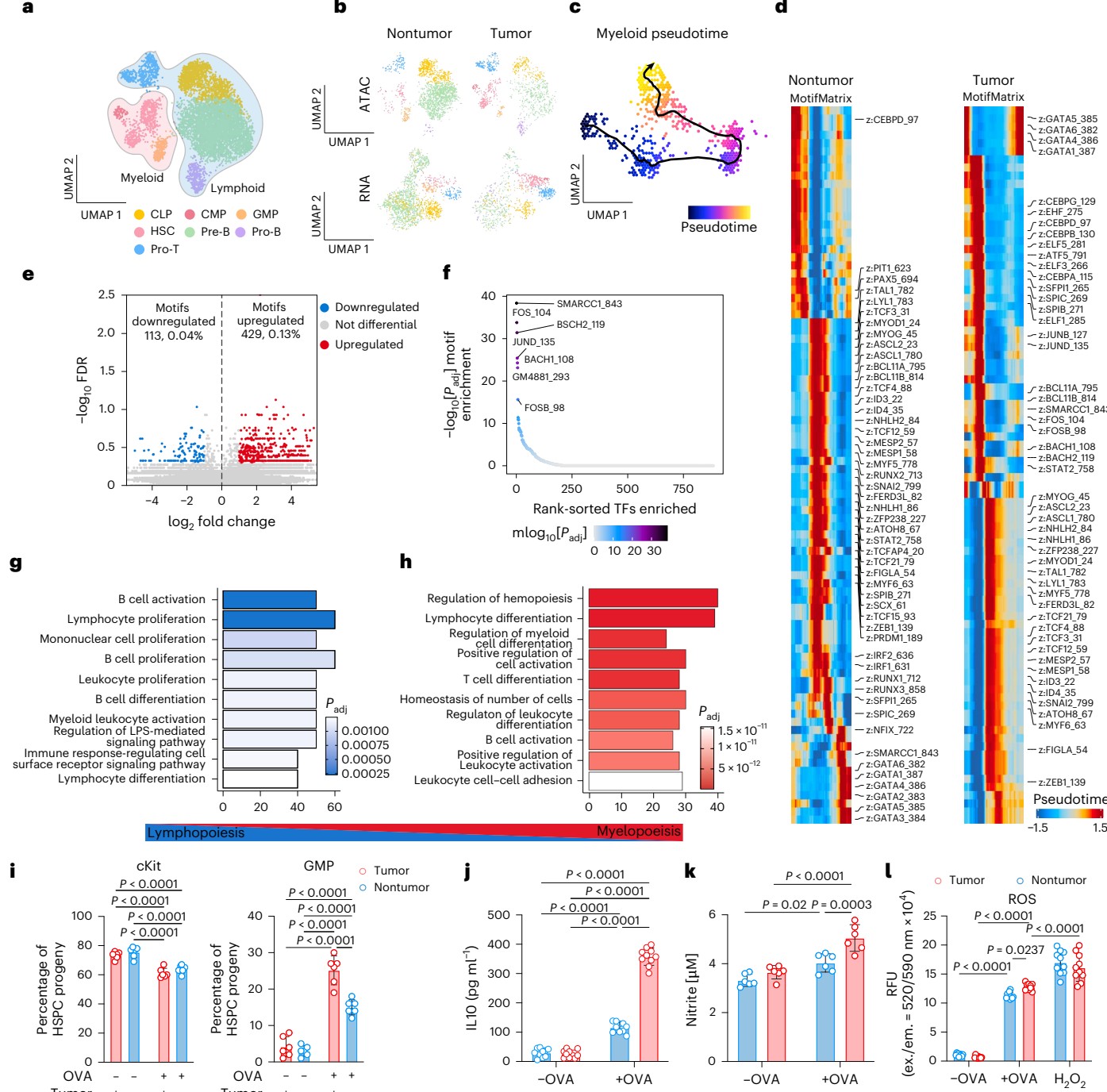

**Fig. 5 | Combined analysis of chromatin accessibility and gene expression in HSCs from skull and tibia of EP^ZFTA-RELA-bearing and control mice reveals myelopoiesis bias in skull HSCs. a**, UMAP visualization of the snATAC dataset (1,623 nuclei from LSK⁺CD34⁺ HSCs sorted from the skull and tibia of EP^ZFTA-RELA-bearing and control mice), colored by cluster: CLP, CMP, GMP, HSC, pre-B cell, pro-B cell and pro-T cell. **b**, UMAP separated by genotype and modality. **c**, ArchR pseudotime visualization of the differentiation trajectory of hematopoietic cells from the skull of EP^ZFTA-RELA-bearing mice. **d**, Heatmap of motifs identified across the myeloid cell trajectory, with ArchR split between control skull and tumor skull. **e**, Motif enrichment in differential peaks upregulated in skull of

tumor-bearing relative to control mice, visualized by volcano plot. **f**, Significant rank-sorted transcription factor motifs enriched in skull HSCs of tumor-bearing mice. **g,h**, Top downregulated (**g**) and upregulated (**h**) genes from gene ontology pathway analysis in skull-derived HSCs relative to those from tibia. **i**, Indicated populations derived from ex vivo OVA-primed HSPCs following coculture with OT-II CD4⁺ T cells (n = 6; mean ± s.e.m.; one-way ANOVA with Šídák's test). **j–l**, Cytometric bead array measurements of IL-10 production (**j**), nitrite production (**k**) and ROS production (**l**) in OVA-primed HSPC-derived populations following coculture with OT-II CD4⁺ T cells (n = 10; mean ± s.e.m.; one-way ANOVA with Šídák's test). ex., excitation; em., emission.

hematopoiesis remodels local immune niches to favor antitumor immunity and demonstrate on-target activity of intravenous mavrilimumab in combination with checkpoint blockade, underscoring its translational potential.

Together, these data identify the skull marrow as a druggable immunological niche in children with a brain tumor and potentiate clinical exploration of mavrilimumab in combination with immune checkpoint blockade therapies such as anti-CTLA-4 for childhood brain tumors.

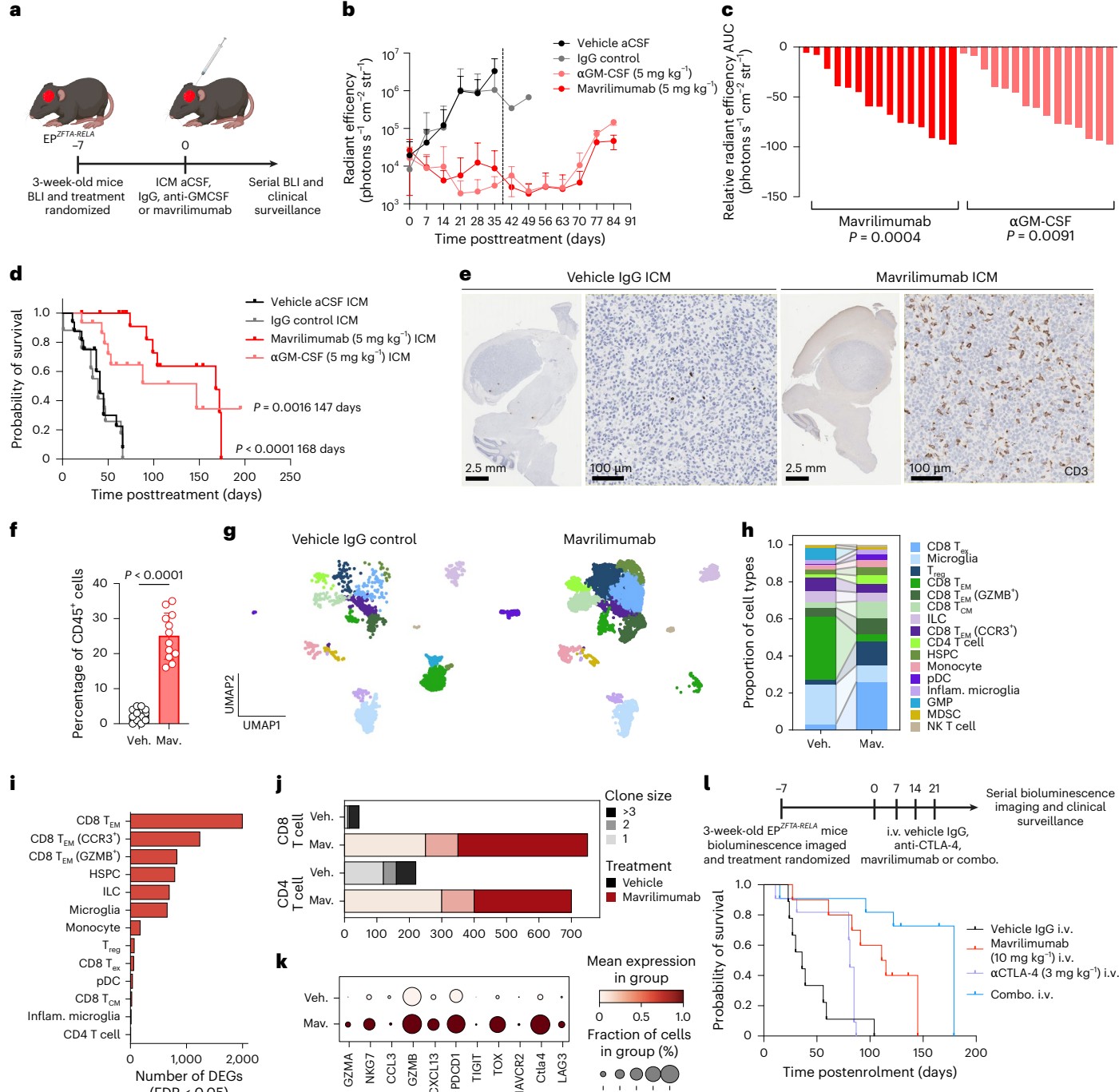

**Fig. 6 | Normalizing skull hematopoiesis improves survival of *ZFTA–RELA*-fusion-driven ependymoma-bearing mice. a**, Experimental design for the treatment of 3-week-old *ZFTA–RELA* EP$^{ZFTA-RELA}$-bearing mice with a single ICM injection of 10 µl of vehicle aCSF ($n = 17$), IgG isotype control (5 mg kg$^{-1}$, $n = 16$), anti-GM-CSF (5 mg kg$^{-1}$, $n = 14$) or mavrilimumab (5 mg kg$^{-1}$, $n = 17$). **b**, Weekly bioluminescence tracking of treated mice; data represent the mean ± s.d. **c**, Waterfall plots of the area under the curve for anti-GM-CSF ($P = 0.0004$, $n = 15$) and mavrilimumab-treated mice ($P = 0.0091$, $n = 16$) up to 42 days; Welch's $t$-test. **d**, Kaplan–Meier survival plot of EP$^{ZFTA-RELA}$-treated mice; log-rank Mantel Cox test. **e**, Representative immunohistochemical images of brains from mice treated with vehicle or mavrilimumab at recurrence. **f**, Flow cytometry quantification of CD8 T cells as a proportion of CD45$^+$ cells; $n = 12$ mice per group; data represent the mean ± s.d. **g**, UMAP visualization of scRNA-seq of extravascular CD45$^+$ cells isolated from the tumor parenchyma of vehicle or mavrilimumab-treated mice at recurrence, colored by cell type: exhausted CD8 T cell, microglia, CD8 tissue effector memory, CD8 T$_{EM}$ GZMB$^+$, CD8 T$_{CM}$, CD4 T cell, HSPC, monocyte,

plasmacytoid, inflammatory microglia, GMP, MDSC and natural killer T cell. **h**, Quantification of proportions of cell types in each group. **i**, Quantification of numbers of differentially expressed genes in each cell type in vehicle-treated relative to mavrilimumab-treated mice. **j**, Single-cell TCR sequencing clonotype analysis of numbers of expanded clones in CD4$^+$ and CD8$^+$ T cells in vehicle-treated and mavrilimumab-treated mice ($n = 6$ to 10 tumors pooled per group). **k**, Dot plot of average and percentage expression of exhaustion markers in CD8 exhausted T cell population in vehicle-treated and mavrilimumab-treated mice. **l**, Protocol and Kaplan–Meier survival plot for EP$^{ZFTA-RELA}$-treated mice, with four weekly intravenous injections of IgG control ($n = 9$), mavrilimumab (10 mg kg$^{-1}$, $n = 10$), anti-CTLA-4 (3 mg kg$^{-1}$, $n = 11$) or combination treatment (mavrilimumab + anti-CTLA-4, $n = 11$); log-rank Mantel Cox test. AUC, area under the curve; combo., combination treatment; DEGs, differentially expressed genes; mav., mavrilimumab; pDC, plasmacytoid dendritic cell; T$_{ex}$, exhausted T cell; veh., vehicle. Illustrations in **a** created with BioRender.com.

## Discussion

Skull bone marrow, once viewed as a hematopoietic niche, is now recognized as an active immunological interface within the CNS[7,13,27]. We show that tumor-borne CSF cues instruct role skull bone marrow hematopoiesis, revealing a population of HSPCs with antigen-presentation capacity that mirrors counterparts previously noted in the peripheral bone marrow[25]. In EP$^{ZFTA-RELA}$, tumor-derived antigens and cytokines drive these HSPCs to engage CD4$^+$ T cells, promoting myeloid-biased differentiation and T$_{reg}$ cell polarization and contributing to tumor immunotolerance. This circuit, likely adapted to limit neuroinflammation and self-reactivity, is subverted by childhood brain tumors to maintain immune privilege.

Tumor exposure remodels the chromatin and transcriptional landscape of local HSPCs and expands an MDSC-like lineage, integrating cues across the CSF, dura and skull bone marrow. These findings extend emerging evidence that the CNS border tissues are major hubs of immune education[13,42]. In line with recent observations that FOXP3$^+$CD4$^+$ T$_{reg}$ cells are readily detectable in the dura mater of homeostatic mice and that their ablation leads to enrichment of IFNγ-producing cells, we demonstrate that polarization of CD4 T cells towards T$_{reg}$ cells can occur in the skull bone marrow and dura in response to endogenous CNS and neoantigens[43]. We extend our observations from studies of exogenous peptides to endogenous CNS peptides, with tumor-derived neoantigens, to demonstrate that tumor neoantigens are recognized as self and drive these processes in brain tumors. Together, these data extend the notion that the dura mater serves as a critical site for immunotolerance, favoring immunosuppression, which is subverted in the context of childhood brain tumors. Further work will be required to define precisely which peptides are active in tolerance of patients with childhood EP$^{ZFTA-RELA}$ and how these findings may be implicated in vaccine- and peptide-based immunotherapeutics.

Targeting this axis through intrathecal or intravenous blockade of GM-CSF or GM-CSFRα disrupted HSPC-driven myelopoiesis, reduced intratumoral MDSCs and induced profound tumor suppression, consistent with CD8 T cell expansion, with synergy observed alongside CTLA-4 blockade. These data suggest that cytokine-directed modulation of skull bone marrow hematopoiesis could reveal a therapeutic vulnerability in childhood brain tumors, while avoiding the toxicities of current treatments.

In summary, we identify a new therapeutic vulnerability in childhood brain tumors, in which the tumor educates the local immune supply from the skull bone marrow at the apex of hematopoiesis. Our findings demonstrate that CNS-derived signals can regulate skull hematopoiesis to enforce a locally tolerant immune repertoire, fostering an immunosuppressive environment conducive to tumor development.

## Online content

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

¹CRUK Cambridge Institute, University of Cambridge, Li Ka Shing Centre, Cambridge, UK. ²Department of Medicine, University of Cambridge, Addenbrooke's Hospital, Cambridge, UK. ³Cambridge Institute of Therapeutic Immunology and Infectious Diseases, University of Cambridge, Cambridge, UK. ⁴Genetics Branch, NCI, NIH, Bethesda, MD, USA. ⁵Academic Neurosurgery Division, Department of Clinical Neurosciences, University of Cambridge, Cambridge, UK. ⁶The Eli and Edythe Broad Center of Regeneration Medicine and Stem Cell Research, University of California, San Francisco, San Francisco, CA, USA. ⁷Faculty of Health, Medical and Behavioral Sciences, Frazer Institute, University of Queensland, Brisbane, Queensland, Australia. ⁸Department of Immunology, Genetics and Pathology, Science for Life Laboratory, Rudbeck Laboratory, Uppsala University, Uppsala, Sweden. ⁹Department of Oncology, Cambridge Biomedical Campus, Cambridge, UK. ✉e-mail: Elizabeth.Cooper@cruk.cam.ac.uk; Richard.Gilbertson@cruk.cam.ac.uk

## Methods

### Experimental model and participant and subject details

**Human participants.** Human brain tissue was obtained from patients undergoing neurosurgical procedures at Cambridge University Hospitals NHS Foundation Trust. Tissue collection and use for research were approved by the East of England–Cambridge Central Research Ethics Committee (REC reference: 23/EE/0241). Written informed consent was obtained from all participants before inclusion, in accordance with the Declaration of Helsinki.

**Mice.** All animal work was carried out under the Animals (Scientific Procedures) Act 1986 in accordance with the UK Home Office license (project license PP9742216) and approved by the Cancer Research UK Cambridge Institute Animal Welfare and Ethical Review Board. Mice were housed in individually ventilated cages with wood chip bedding plus cardboard fun tunnels and chew blocks under a 12 h light/dark cycle at 21 ± 2 °C and 55% ± 10% humidity. A standard diet was provided with ad libitum water. Mice were allowed to acclimate for at least 1 week in the animal facility before the beginning of any experiment. Adult males and females between 4 and 6 weeks of age were primarily used for our studies unless stated otherwise. Sample sizes were determined on the basis of a power analysis in accordance with previously published experiments. Experimenters, where necessary, were blinded to experimental groups during both scoring and quantification. Mouse strains used are listed in Supplementary Table 1.

### Method details

**Generation of *Rosa26*-locus-targeted conditional, Nestin-driven, *C11orf95–RelA* fusion expressing mice.** A conditional *C11orf95:RelA* fusion (*Nestin-(lsl)-C11orf95:RelA*) construct with homology to the *Rosa26* locus was generated by conventional molecular techniques. The targeting plasmid was linearized and gel purified before being nucleofected into C57BL/6J embryonic stem cells. After G418 sulfate selection, clones were isolated and subjected to genotyping and karyotyping. Four correctly targeted clones were injected into wild-type CD1 eight-cell-stage embryos. The microinjected embryos were cultured in KSOM-AA (KCl, enriched simplex optimization medium with amino acid supplement, Zenith Biotech) at 37 °C with 95% humidity and 5% $CO_2$ until the blastocyst stage and then transferred into pseudopregnant recipients. The resulting $F_0$ mice were bred to C57BL/6J mice, proving germline transmission and establishing the colony (see Supplementary Table 1 for details of animals and Supplementary Table 4 for oligonucleotide sequences).

**Orthotopic allotransplantation models.** CPC and group 3 medulloblastoma orthotopic allotransplantation models were generated from in vitro cultures of cells generated from genetically engineered mouse models of each of these tumors. Briefly, as an extension of our previous work[28], CPC cell lines were derived from *Pten⁻ᐟ⁻Rb⁻ᐟ⁻TpS3⁻ᐟ⁻* mice donated by S. J. Baker[44] crossed with TTR *CreEsr1;TdT* mice (MGI: 3046546)[45]. Cell lines derived from these tumors were expanded in neurobasal medium with B27 without vitamin A, N2 supplement, and rEGF, FGFb and IGF-2 (20 ng ml⁻¹) on extracellular-matrix-coated flasks. Group 3 medulloblastoma lines were donated by F.J.S.[29] and were cultured in neurobasal medium with B27 without vitamin A, N2 supplement, and rEGF, FGFb and IGF-2 (20 ng ml⁻¹). Both lines were transfected with luciferase (pLenti CMV Puro LUC (w168-1)) before implantation. P1 C57B/L6 mice were orthotopically implanted with 400 cells (G3 medulloblastoma) or 4,000 cells (CPC) on the right temporal lobe, or lateral ventricle, respectively, under anesthesia.

**In vivo preclinical studies.** EP*ᶻᶠᵀᴬ⁻ᴿᴱᴸᴬ*, CPC and Gr3 medulloblastoma tumor-bearing mice were randomized at 3 weeks old, confirmed by bioluminescence exploiting the endogenous *Nestin*^Cre-ZFTA-RELA^–*IRES*–luciferase allele in EP*ᶻᶠᵀᴬ⁻ᴿᴱᴸᴬ* or lentiviral introduction of luciferase

in vitro in CPC and Gr3 medulloblastoma as described above. Mice received a single intrathecal injection of mavrilimumab (5 mg kg⁻¹; an anti-GM-CSF receptor alpha antibody) or control antibody and were monitored with biweekly bioluminescence imaging to monitor tumor burden until mice reached a humane endpoint based on accumulation of clinical signs. For intravenous administration, EP*ᶻᶠᵀᴬ⁻ᴿᴱᴸᴬ* bearing mice were randomized at 5 weeks old to receive mavrilimumab (10 mg kg⁻¹), alone or with anti-CTLA-4 (3 mg kg⁻¹) or control antibody, once weekly for 4 weeks.

**Tissue processing and immunohistochemistry.** Meningeal whole mounts were prepared as previously described[36]. Briefly, mice were given a lethal dose of anesthesia via intraperitoneal injection (Dolethal 10% v/v) and transcardially perfused with 0.025% heparin phosphate-buffered saline (PBS). Mice were decapitated posterior to the occipital bone and, following removal of overlying skin and muscle from the skull, the skull cap was removed and drop-fixed in paraformaldehyde (4% w/v in PBS) for 1 h at 4 °C, and the dura mater was carefully peeled and washed. Whole mounts were blocked and permeabilized in buffer containing 10% Tris, 1% bovine serum albumin (BSA), 1% serum, 1% saponin and 0.5% Triton X-100 for 1 h at room temperature. Samples were incubated with primary antibodies in blocking buffer overnight at 4 °C, washed and, if required, incubated with secondary antibodies for 2 h at room temperature. Prepared dura were mounted using Fluoromount-G for imaging.

**Confocal microscopy and image analysis.** Whole-mount meninges were imaged using a Leica Stellaris SP8 or STELLARIS 8 confocal microscope with multiple laser lines and objectives (×4–60). Tile scans were imported into Imaris v.9.5 (Bitplane) for three-dimensional analysis. Regions of interest, including perisinus and cortical areas, were manually segmented using the surface function. Absolute cell numbers were quantified by thresholding of positively stained cells within each surface using the spot-detection function, and statistics were exported for analysis. For histocytometry, surfaces encompassing the whole meningeal mount were generated for quantification of fluorescence intensity, cell frequency and spatial distribution. Channel statistics were exported to Excel and converted to FCS format for visualization in FlowJo (TreeStar).

**Sample processing for scRNA-seq.** Age-matched 6- to 8-week-old EP*ᶻᶠᵀᴬ⁻ᴿᴱᴸᴬ* and Nestin*^CreERT2^* mice were intravenously injected with CD45-PE 3 min before schedule 1. Blood was collected via retro-orbital sampling, and red blood cell lysis was performed by resuspension in 1 ml of ACK lysis buffer (Quality Biological). The pellet was resuspended in FACS buffer (0.1 M, pH 7.4 PBS with 1% BSA and 1 mM EDTA) until use. Meningeal dura was carefully collected under a dissection microscope. Meninges and calvaria were then digested for 15 min at 37 °C with constant agitation using 1 ml of prewarmed digestion buffer (Dulbecco's modified Eagle medium (DMEM) with 2% fetal bovine serum (FBS), 1 mg ml⁻¹ collagenase D (Sigma Aldrich) and 0.5 mg ml⁻¹ DNase I (Sigma Aldrich)) and filtered through a 70-μm cell strainer, and enzymes were neutralized with 1 ml of complete medium (DMEM with 10% FBS). For peripheral bone marrow, both tibia were flushed with 0.05% BSA PBS with 0.05% EDTA; the resulting samples were filtered through 100-μm filters, washed with 2% FBS in RPMI and resuspended in 0.05% BSA PBS solution. Whole intact deep cervical lymph nodes were mashed through a 70-μm cell strainer using a sterile syringe plunger and washed with 5 ml of FACS buffer. Tumor and brain samples were macrodissected, with removal of choroid, and dissociated using a mouse tumor dissociation kit (Miltenyi Biotec) with a gentleMACS Octo Dissociator (Miltenyi Biotec). Samples were resuspended in 40% Percoll and centrifuged at 600*g* for 10 min. Supernatant was removed, washed with 2% FBS in RPMI, and resuspended in 0.05% BSA PBS solution. Samples were stained with DAPI (0.2 μg ml⁻¹). Samples then were centrifuged,

resuspended in FACS buffer with anti-CD16/32 ($F_c$ block, BioLegend; diluted 1:50 in the FACS buffer), and incubated with fluorescently conjugated antibodies (anti-CD45 APC and anti-Ter119 FITC) at 4 °C for 20 min. Cells were sorted using an Influx Cell Sorter (BD Biosciences) or FACSAria II (BD Biosciences) into 1% BSA-coated 1.5-ml Eppendorf tubes with 500 µl of DMEM.

**Sample processing for single-nucleus RNA-seq.** Single-cell suspensions of skull and tibial bone marrow cells were obtained as described above. Cells were sorted for live CD45 i.v.⁻Ter119⁻CD45⁺Lin⁻CD34⁺ using an Influx Cell Sorter (BD Biosciences) or FACSAria II (BD Biosciences) into 1% BSA-coated 1.5-ml Eppendorf tubes with 500 µl of DMEM.

FACS-isolated cells were centrifuged at $500g$ for 5 min at 4 °C before removal of the supernatant and resuspension in 500 µl lysis buffer (10 mM Tris-HCl, 3 mM $MgCl_2$, 2 mM NaCl, 0.005% NP-40 substitute, 0.1 mM DTT, SUPERase RNase inhibitor 0.25 U ml⁻¹ and protease inhibitor (A32965)) for 2 min. Nuclei were pelleted at $500g$ for 5 min at 4 °C, washed in PBS with 1% BSA and counted using Trypan Blue exclusion. Single-nucleus multiome libraries (RNA + ATAC) were prepared using a Chromium Next GEM Single Cell Multiome ATAC + Gene Expression kit (10x Genomics) according to the manufacturer's protocol. Libraries were sequenced on an Illumina NovaSeq 6000.

**Sample processing of human neurosurgical tissue.** To obtain flow cytometry of skull, dura and tumor shown in Supplementary Fig. 4, discard neurosurgical material from skull fragments, dura and tumor was collected from a 6-month-old male patient undergoing routine tumor debulking for an atypical choroid plexus papilloma (WHO grade 2). Skull fragments and dura obtained during craniotomy were placed in RPMI-1640 supplemented with 10% FBS and kept on ice before being dissociated as described for mouse tissues. Debris was removed using MACS debris removal solution (130-109-398) before flow cytometry.

Freshly resected human brain tumor tissue was dissected to remove cauterized regions, cut into ~3–8 mm³ pieces and rinsed in HBSS. Tissue was enzymatically and mechanically dissociated using a gentleMACS Octo Dissociator with Heaters (Miltenyi Biotec) according to the manufacturer's instructions. Cell suspensions were filtered through 40-µm strainers and pelleted, and myelin was removed via a 40% Percoll gradient. Resulting cell pellets were resuspended in PBS for downstream flow cytometry. Tissue collection was performed under Cambridge University Hospital REC approval (23/EE/024).

**Flow cytometry.** Cell suspensions were prepared as described above and transferred into a V-bottomed plate. Viability staining was performed using Zombie NIR (1:500 in PBS, 10 min, room temperature; BioLegend). Suspensions were then pelleted ($450g$ for 5 min) and resuspended in anti-CD16/32 antibody (1:100, BioLegend) diluted in FACS buffer to block $F_c$ receptor binding. Antibodies against cell surface epitopes were then added for 10 min at room temperature. For a full list of antibodies, see Supplementary Table 3. Flow cytometry was performed using an Aurora spectral flow cytometer (Cytek Biosciences), and data were analyzed with FlowJo (v.10, BD Biosciences).

**CSF collection and intracisterna magna injections.** Mice were anesthetized via intraperitoneal injection of ketamine (100 mg kg⁻¹) and xylazine (10 mg kg⁻¹) in saline and placed on a stereotactic frame. The fur over the incision site was clipped, and the skin was disinfected with three alternating washes of alcohol and Betadine. For intracisterna magna injections, a midline incision was made, and the posterior nuchal musculature was divided, exposing the inferior, dorsal aspect of the occipital bone and the posterior dura overlying the cisterna magna. A glass capillary attached to a microinjector (World Precision Instruments) was used. A 5-µl volume was infused, and injection rates were adjusted to achieve a 5-min injection, followed by a 5-min wait period to prevent backflow. For CSF collection, a glass capillary was inserted

through the dorsal dura mater into the superficial cisterna magna, and approximately 50 µl of CSF was drawn by capillary action. For CSF transfer experiments, 10 µl of CSF was transferred at a rate of 400 nl min⁻¹.

**CSF collection and multiplex analyte analysis.** Mice were anesthetized via intraperitoneal administration of ketamine and xylazine and placed on a stereotactic frame. CSF was collected from the cisterna magna with a 30-gauge needle. CSF (12.5 µl) was obtained from each mouse, and analytes were quantified using Luminex magnetic beads according to the Bio-Plex Pro Mouse Cytokine Panel 23-plex instructions (Bio-Rad). Data were acquired with a Luminex FLEXMAP 3D and analyzed with xPONENT software (v.4.2, Luminex).

**Sample dissolution, TMT labeling and reverse-phase fractionation.** Thirty-milliliter aliquots of CSF were lysed in 100 mM TEAB, 10% isopropanol, 50 mM NaCl and 1% sodium deoxycholate with nuclease and protease/phosphatase inhibitors, incubated for 15 min at room temperature and sonicated. Samples were reduced with TCEP and alkylated with iodoacetamide (1 h each), then digested overnight at 37 °C with trypsin (1:30). Dried peptides were resuspended in 0.1 M TEAB and labeled with TMTpro reagents for 1 h. A premix was analyzed to normalize sample inputs before quenching with hydroxylamine. Combined samples were acidified to remove sodium deoxycholate and fractionated at high pH on reverse-phase cartridges into nine fractions (5–50% acetonitrile), then dried and reconstituted in 0.1% formic acid.

**Liquid chromatography–tandem mass spectrometry.** Peptide fractions were analyzed on a Vanquish Neo UHPLC system coupled with an Orbitrap Ascend (Thermo Scientific) mass spectrometer. Peptides were trapped on a 100 µm ID × 2 cm microcapillary C18 column (5 µm, 100 A), followed by 90 min elution using a 75 µm ID × 25 cm C18 RP column (3 µm, 100 A) at a flow rate of 300 nl min⁻¹. A real time search (RTS)–MS3 method was used for the analysis; MS1 spectra were acquired in the Orbitrap ($R$ = 120 K; scan range: 400–1,600 $m/z$; AGC target = 400,000; maximum IT = 251 ms) and MS2 spectra in the ion trap (isolation window: 0.7 Th; collision energy = 30%; maximum IT = 35 ms; centroid data). For RTS, trypsin/P digestion was selected using static cysteine carbamidomethylation and TMTpro modification on lysine and peptide amino terminus. The search was conducted for a maximum of 35 ms with the following thresholds: Xcorr = 1.4, dCn = 0.1, precursor ppm 10, charge state = 2. MS3 spectra were collected in the Orbitrap (R = 45 K; scan range: 100–500 $m/z$; normalized AGC target = 200%; maximum IT = 200 ms; centroid data). Phospho fractions were subjected to MS2 analysis without RTS.

**Data processing.** Spectra were processed in Proteome Discoverer 3.0 using SequestHT against the UniProt mouse database. Static modifications included TMTpro (N termini, lysine) carbamidomethyl at cysteines (+57.021 Da). Methionine oxidation (+15.9949 Da) and deamidation (+0.984) on asparagine were included as dynamic modifications. Searches used 20 ppm precursor and 0.5 Da fragment tolerances. Peptides were filtered to 1% false discovery rate (FDR), and unique peptides were used for quantification.

**Single-cell library construction and sequencing.** For scRNA-seq, gel bead-in-emulsions were prepared by loading up to 10,000 cells per sample onto a Chromium Chip G (10x Genomics, 1000073) and run using a Chromium Controller (10x Genomics). Complementary DNA (cDNA) libraries were generated using a Chromium Single Cell 5′ GEM, Library and Gel Bead Kit V3 (10x Genomics) with V(D)J. Libraries were sequenced using a NovaSeq 6000 Kit (v.2.5, Illumina) on an Illumina NovaSeq 6000 system.

**scRNA-seq analysis.** Cell Ranger (v.7.1.0) outputs were processed using Scanpy (v.1.8.2). Doublets were identified with Scrublet (.v0.2.3) using an iterative subclustering approach as previously described,

computing median scrublet scores per subcluster and flagging outliers by one-tailed $t$-test with Benjamini–Hochberg correction (FDR < 0.1). Quality control followed the sc-dandelion preprocessing pipeline (max_genes = 6,000; min_genes = 200; GMM-based mitochondrial threshold). Genes expressed in ≥3 cells were retained, and counts were normalized to 10,000 unique molecular identifiers per cell. Highly variable genes were selected using Scanpy defaults (min_mean = 0.0125, max_mean = 3, min_disp = 0.5). Principal component analysis (PCA) was followed by batch correction with harmonypy (v.0.0.5). Clustering used Leiden (v.0.8.2; resolution = 1.0), and UMAP (v.0.5.1; min_dist = 0.3) for visualization. Cell types were annotated by canonical markers and Wilcoxon rank-sum statistics.

**Differential expression and enrichment analysis.** Subsets were defined by cluster identity, retaining genes with ≥4 counts in ≥4 cells. Highly variable genes ($n = 2,000$) were identified using SingleCellExperiment (v.3.20) with modelGeneVar and getTopHVGs (scran, v.1.35.0). Differential expression was determined using limma (v.3.2.0) and edgeR (v.4.0) with lmFit, contrasts.fit and eBayes; genes with FDR < 0.05 were considered to be significantly differentially expressed. Gene ontology overrepresentation analyses were performed using clusterProfiler (v.3.2.0) with enrichGO (gene set size: 10–500; Benjamini–Hochberg-adjusted $P < 0.05$).

**Ligand–receptor analysis.** Proteins detected by CSF liquid chromatography–mass spectrometry (Supplementary Table 16) were mapped to gene IDs using biomaRt (v.3.20). Ligands were filtered using RNAMagnet (v.0.1.0; getLigandsReceptors, cellularCompartment = "secreted", "ECM" or "both"; version = 3.0.0)[46]. Corresponding receptors (Supplementary Table 17) were visualized as average normalized expression using circlize (v.0.4.16).

**Cytokine signaling inference.** Cytokine activity was assessed using CytoSig, which was applied to log-normalized, scaled and batch-corrected matrices[47]. Activity scores were derived from perturbation-informed cytokine signatures and aggregated across cell types to compare tumor and control tissue microenvironments.

**TCR sequencing analysis.** V(D)J libraries were processed using Cell Ranger (v.7.1.0). Clonotypes were analyzed with Dandelion (v.0.1.5), retaining high-confidence, productive αβ pairs. Clonal expansion, diversity, overlap and integration with transcriptomic metadata were computed using Dandelion's built-in functions[48].

**scRNA-seq integration and analysis of human data.** Cell Ranger (v.7.0) outputs were processed with Scanpy (v.1.8.2). Doublets were detected with Scrublet (v.0.2.3) using an iterative subclustering approach, computing median cluster Scrublet scores and flagging outliers by one-tailed $t$-test with Benjamini–Hochberg correction (FDR < 0.1). QC followed the sc-dandelion workflow (max_genes = 6,000; min_genes = 200; GMM-derived mitochondrial threshold). Genes expressed in <3 cells were excluded. Counts were normalized to 10,000 per cell, yielding 125,000 high-quality cells.

Highly variable genes were selected (≤3 mean ≥ 0.0125 and dispersion ≥ 0.5), followed by log-transformation, scaling, PCA and batch correction with harmony (v.0.0.5). Leiden clustering (v.0.8.2; resolution = 1.0; neighborhood size = 10) and UMAP (v.0.5.1; min_dist = 0.3) were used for dimensionality reduction. Cell identities were assigned by canonical markers and Wilcoxon rank-sum testing.

For cross-dataset integration, batch and confounder effects were regressed using Ridge regression (scikit-learn v.1.3.0), and corrected principal components were used as input to BBKNN to generate a batch-balanced graph. UMAP and Leiden clustering on this graph defined shared cell states across scRNA-seq and single-nucleus RNA-seq datasets.

**snATAC + GEX multiome analysis.** Libraries were sequenced on a NovaSeq 6000 to a minimum depth of 25,000 unique fragments per nucleus. Raw data were processed with Cell Ranger ARC (v.2.0). Fragment files were imported into R (v.4.4.0) using Signac and Seurat. QC metrics (total fragments, fraction of reads in peaks, nucleosome signal) were computed; nuclei with <1,000 fragments or nucleosome signal >2 were removed. Doublets were identified with DoubletFinder (v.2.0.4).

Peaks were called on aggregated ATAC data using MACS2 (v.2.2.9.1) and used to construct a peak-by-cell matrix. Dimensionality reduction was performed using latent semantic indexing followed by PCA, and UMAP was used for visualization. For paired RNA–ATAC datasets, multimodal integration was performed in Seurat (v.4.4.0) using FindMultiModalNeighbors to generate a shared UMAP embedding and joint clusters.

Differentially accessible regions were identified using ArchR's getMarkerFeatures() (Wilcoxon rank-sum). Motif enrichment was performed using chromVAR (v.1.16.0) with cisBP annotations. Pseudotime trajectories were inferred with ArchR's addTrajectory() to model dynamic chromatin changes. Analyses were performed in R (v.4.3.1) with visualization using ggplot2 (v.3.4.3) and patchwork (v.1.1.3).

**Quantitative PCR.** Cells were sorted directly into RNA lysis buffer (Arcturus PicoPure) and processed for cDNA synthesis using SuperScript VI (Invitrogen). cDNA was diluted 1:10 and amplified in technical triplicates using PowerUP SYBR Green Master Mix (Thermo Fisher) with intron-spanning primers (Supplementary Table 4) on a QuantStudio 7 (Applied Biosystems). Cycling was as follows: 50 °C for 2 min, 95 °C for 10 min, 40 cycles of 95 °C for 15 s and 60 °C for 1 min. Relative gene expression was normalized to that of $Gapdh$ and $Actb$ and calculated as $2^{-\Delta Ct}$, where ΔCt = (geometric mean of housekeeper Ct) − (gene of interest Ct).

**Bone marrow transplantation.** To test the stem cell potential of intratumoral MHC-II populations, equal numbers ($5 \times 10^4$) of CD45[+]Lin[−]MHC-II[+] or MHC-II[−] intratumoral cells were transplanted intravenously into mice that had undergone nonmyeloablative conditioning with busulfan (25 mg kg[−1]). Sixteen weeks posttransplantation, total bone marrow cells ($1 \times 10^6$) were transplanted into secondary recipients. Mice were bled every 4 weeks, and cells were stained as described above to assess engraftments.

**In vitro expansion of naive CD4+ T cells.** Cryopreserved C57BL/6 or 2D2 splenocytes were thawed using standard protocols and washed in FACS buffer (PBS with 2% FBS and 2 mM EDTA) and isolated using magnetic beads (Miltenyi Biotec). Negative fractions were collected during column loading, and positively selected T cells were eluted by removal of the column from the magnet and flushing with buffer using a plunger.

Viable CD4+ T cells were counted and cultured in AIM-V medium supplemented with 5% heat-inactivated FBS, penicillin–streptomycin, L-glutamine, and recombinant mouse IL-2 (40 IU ml[−1]; R&D Systems). Cells were maintained at a density less than $1.5 \times 10^6$ cells ml[−1] and stimulated with Dynabeads Mouse T-Expander CD3/CD28 beads ($1 \times 10^8$ per ml; Invitrogen, 111.41D) following the manufacturer's protocol.

**Adoptive transfer and immunization.** To test the ability of endogenous CNS antigens to polarize skull CD4 T cells, we adoptively transferred expanded naive CD4+FOXP3−CD90.1+ T cells into RAG2[−/−] recipient mice. Mice were immunized intrathecally with aCSF, MOG$_{35-55}$ peptide or a MOG-based fusion peptide at days 21 and 42 posttransfer. At day 49, dura and skull bone marrow tissues were harvested and analyzed for FOXP3 expression within the transferred CD90.1+ T cell population by flow cytometry.

To test the ability of endogenous CNS antigens to polarize antigen-specific CD4 T cells, we expanded naive CD4+ T cells ($5 \times 10^4$) from

2D2 TCR transgenic mice in vitro and transferred them into wild-type recipient mice. Mice received intrathecal injections of aCSF or $MOG_{35-55}$ peptide (2.5 µl of 2 mg ml$^{-1}$) on days 1 and 7. On day 14, dura and skull bone marrow were harvested and analyzed for $FOXP3^+CD4^+$ T cells.

**In vivo antigen presentation assays.** For analysis of presentation of exogenous antigens on HSPCs, DQ-OVA (100 µg ml$^{-1}$, Invitrogen), Eα peptide (52–68) (500 µg ml$^{-1}$, Mimotopes) or a control IgG2b antibody (100 µg ml$^{-1}$, eB149/10H5, Thermo Fisher) were injected intrathecally as described above, and mice were sacrificed after 6 h.

**Murine ex vivo cultures.** Cells were cultured at 37 °C and 5% $CO_2$ in U-bottomed plates in a total volume of 200 µl of DMEM GlutaMAX (Gibco) supplemented with 10% heat-inactivated fetal calf serum (Gibco), sodium pyruvate (1.5 mM, Gibco), L-glutamine (2 mM, Gibco), L-arginine (1×, Sigma), L-asparagine (1×, Sigma), penicillin/streptomycin (100 U ml$^{-1}$, Sigma), folic acid (14 µM, Sigma), MEM nonessential amino acids (1×, Thermo Fisher), MEM vitamin solution (1×, Thermo Fisher) and β-mercaptoethanol (57.2 µM, Sigma). Sorted cells were labeled with CellTrace Violet. Naive $CD4^+$ T cells ($5 \times 10^4$) were cocultured with HSPCs, dendritic cells or $CD8^+$ T cells ($2 \times 10^4$) with or without antigenic peptides (OVA $_{323-339}$, DQ-OVA, Eα$_{52-68}$) or blocking/control antibodies. LSK cells were cultured for 12 h with or without OVA$_{323-339}$ peptide in medium containing TPO and SCF, then cocultured with OT-II $CD4^+$ T cells for 72 h.

**Griess assay and ROS production in LSK progeny.** To quantify levels of nitrite and ROS, LSK cells were isolated by FACS and cultured for 12 h in the presence or absence of OVA peptide (50 µg ml$^{-1}$) in culture medium supplemented with TPO (50 ng ml$^{-1}$, PreproTech) and SCF (50 ng ml$^{-1}$, PreproTech) at 37 °C, 5% $CO_2$. These cells were then cocultured with OT-II $CD4^+$ T cells for 72 h. Cells were washed and incubated with a ROS-specific fluorogenic probe for 30 min at 37 °C in the dark, and fluorescence was measured (excitation/emission: 520/605 nm) using a microplate reader. Cells were subsequently lysed, and $NO_2^-$, as a stable nitric oxide product, was quantified using a Griess Reagent Kit (Abcam, ab234044) according to the manufacturer's instructions.

**IL-10 enzyme-linked immunosorbent assay.** To quantify secretion of IL-10, LSK cells were isolated by FACS and cultured for 12 h in the presence or absence of OVA peptide (50 µg ml$^{-1}$) in culture medium supplemented with TPO (50 ng ml$^{-1}$, PreproTech) and SCF (50 ng ml$^{-1}$, PreproTech) at 37 °C and 5% $CO_2$. These cells were then cocultured with OT-II $CD4^+$ T cells for 72 h. IL-10 levels in samples were quantified using a sandwich enzyme-linked immunosorbent assay kit (for example, Abcam Mouse IL-10 ELISA Kit, ab100697), following the manufacturer's instructions. Optical density was measured at 450 nm using a microplate reader (Clariostar, BMG Labtech).

**Colony-forming unit assay.** To observe hematopoietic colony-forming unit formation, the cell suspension obtained from skull bone marrow was seeded in methylcellulose media: (MethoCult H4230 and MethoCult SF H4236, Stemcell Technologies) according to the manufacturer's protocol. Both media were supplemented with IL-3 (20 ng ml$^{-1}$), IL-6 (20 ng ml$^{-1}$), G-CSF (20 ng ml$^{-1}$), GM-CSF (20 ng ml$^{-1}$), SCF (50 ng ml$^{-1}$) and erythropoietin (3 units ml$^{-1}$). After incubation for 14–16 days at 37 °C with 5% $CO_2$, the colonies were characterized and scored according to their morphology on a ZEISS AX10 inverted microscope (Zeiss).

**Intracalvarial injection.** Skull bone marrow delivery of AMD3100 was performed as previously described[12]. Briefly, mice were anesthetized with a mixture of ketamine (100 mg kg$^{-1}$) and xylazine (10 mg kg$^{-1}$). The head of each mouse was shaved, a skin midline incision was made, and the skull was exposed. The outer periosteal layer was thinned using an electrical drill at five spots on top of the skull bone marrow near the bregma and lambda, without damaging the bone marrow. One microliter of 1 mg ml$^{-1}$ AMD3100 (ab120718, Abcam) or vehicle was applied on each spot for 5 min. The skin was sutured, and mice were sacrificed 24 h later.

**EdU and Ki-67 staining for proliferation analysis.** Mice received two intraperitoneal injections of 10 mg of EdU per kilogram of body weight 24 h apart and were sacrificed 24 h after the final injection. After generation of single-cell suspensions for flow cytometry and surface staining as described earlier, fixation, permeabilization and EdU staining were performed following the manufacturer's instructions (Click-iT Plus EdU Alexa Fluor 488 Flow Cytometry Assay Kit, C10632, Thermo Fisher Scientific). Intracellular staining for Ki-67 and additional staining for PE conjugates was performed for 10 min at room temperature after EdU staining.

## Statistics and reproducibility

No statistical methods were used to recalculate or predetermine study sizes, but these sizes were based on those used in similar experiments previously published. Experiments were blinded, where possible, for at least one of the independent experiments. No data were excluded from analyses. For all experiments, animals from different cages were randomly assigned to different experimental groups. All experiments were replicated in at least two independent experiments of at least five mice per group, and all replication was successful. For all representative images shown, images are representative of at least three independent experiments. Statistical tests for each experiment are provided in the respective figure legends. Data distribution was assumed to be normal, but this was not formally tested. In all cases, measurements were taken from distinct samples. Statistical analysis was performed using Prism (v.10.0, GraphPad Software).

## Reporting summary

Further information on research design is available in the Nature Portfolio Reporting Summary linked to this article.

## Data availability

The GEO accession codes for the FASTQ files and quantified gene counts for single-cell sequencing reported in this paper are GSE28237, GSE82459, GSE300889 and GSE300890. All data are available in the main text or the Supplementary Information. Data were also sourced from the following published accessions: nemo:dat-oii74w and dat-7aycjfr (https://assets.nemoarchive.org/collection/nemo:dat-7aycjfr) (NeMO); and GSE141460, GSE126025, GSE156053, GSE226961 and GSE231860 (GEO). The mass spectrometry proteomics data have been deposited to the ProteomeXchange Consortium via the PRIDE partner repository with dataset identifier PXD058239.

## Code availability

All bulk and single-cell analyses and visualizations were performed using R software (v.4.4.0), R studio (v.2024.04.0) and Python (v.3.10). Details on specific packages are included in Methods. No custom code was used for any part of the data processing or analysis.

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

## Acknowledgements

This work was supported by grants to R.J.G. from the Cancer Research UK (CRUK) Centre, CRUK Children's Brain Tumour Centre of Excellence, CRUK Cambridge Institute Core Award; the Brain Tumour Charity Quest for Cures (QfC_2018_10389) and EC Little Princess Trust in partnership with Children's Cancer and Leukemia Group (CCLGA 2024 25 Cooper); and the Brain Tumour Charity Expanding Theories Award (ET_2024_1 10797). We thank all members of the Gilbertson laboratory for participation in discussions that pushed the current study forward. We also acknowledge the microscopy core within the CRUK Cambridge Institute for their support in the acquisition and analysis of meningeal whole mounts; the flow cytometry core for sorting numerous cell types for various experiments in this project; the proteomics and genomics and bioinformatics cores for performing the scRNA-seq, preprocessing and proteomic analysis; and the genome editing core for generation of the EP$^{ZFTA-RELA}$ mouse model: specifically, A. Russel and C. King generated the targeting construct, and X. Zou and P. Papadopolous targeted the embryonic stem cells and injected these to generate founder mice. We thank the CRUK Cambridge Institute Biological Resource Unit for their support with the mouse work. Finally, we extend our gratitude to the patients, donors and families that made this work possible.

## Author contributions

E.C. conducted the bulk of the experimental procedures. D.A.P., C.Y.C.L., L.H., J.T.T., K.W.R., J.E., O.B., G.N., K.E.M., J.M.V., V.N.R.F. and C.S.D. conducted and/or advised on important experimental procedures. S.B., J.K., R.M., C.C., I.J., T.S., L.W., A.R.K., F.J.S., T.Y.F.H., B.W., M.R.C. and R.J.-G. provided important data and reagents. R.J.G. conceived the research and with E.C. designed the approach and oversaw the research. All authors contributed to the writing of the paper.

## Competing interests

R.J.G. is a paid consultant for AstraZeneca Pharmaceuticals. E.C. and R.J.G. have filed patent applications with the Intellectual Property Office of the United Kingdom (application number 2515917.9) related to the work described here. The other authors declare no competing interests.

## Additional information

**Extended data** is available for this paper at https://doi.org/10.1038/s41588-025-02499-2.

**Correspondence and requests for materials** should be addressed to Elizabeth Cooper or Richard J. Gilbertson.

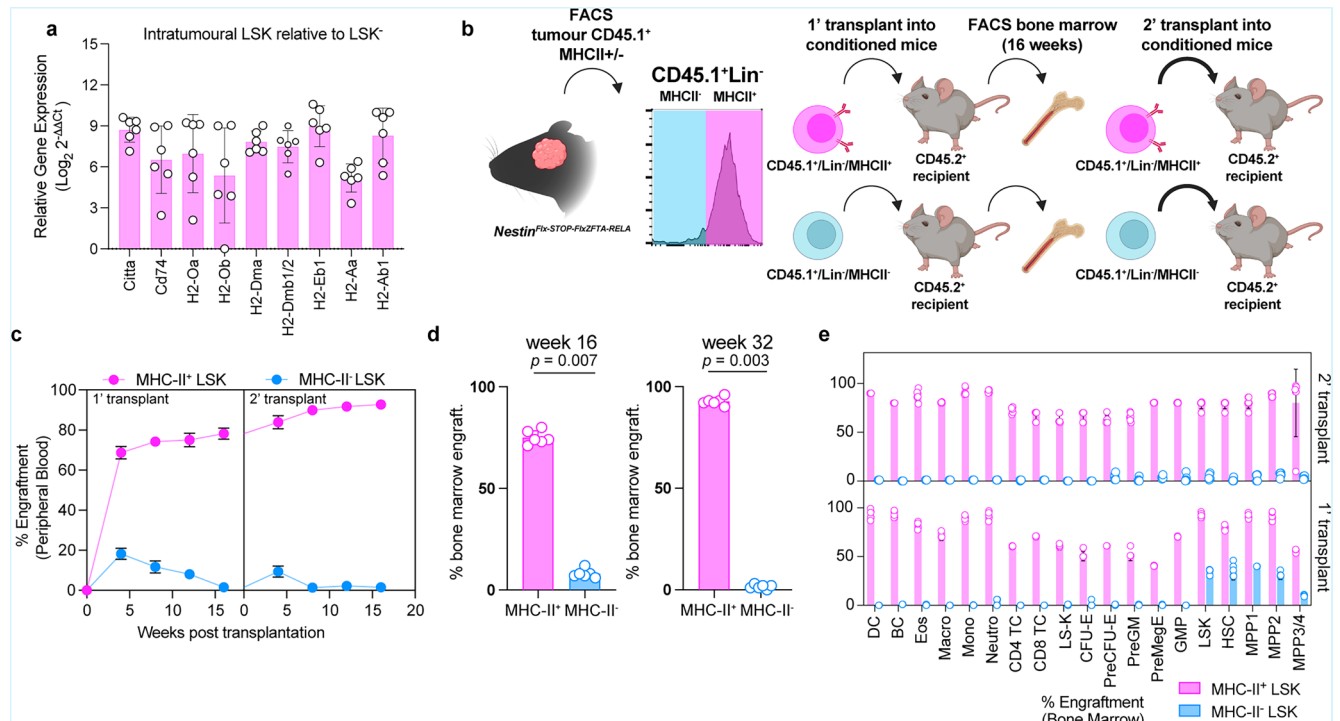

**Extended Data Fig. 1 | Intratumoural pluripotent haematopoietic stem progenitor cells possess antigen-presentation machinery. a**, Messenger ribonucleic acid (mRNA) transcript expression of genes encoding antigen presentation molecules and machinery in fluorescent activated cell-sorting (FACS)-isolated LSK⁺ cells, isolated from EP^ZFTA-RELA tumours, compared to LSK⁻ cells (n = 7, 10 pooled mice per replicate). **b**, Experimental design pertaining to Figures **b**–**e**. **c**, Peripheral blood engraftment of MHC-II Lin- cells transplanted from tumors over 16 weeks (n = 6, 8 pooled mice per replicate) **d**, Quantification of bone marrow engraftment of MHC-II⁺ and MHC-II- tumor-derived Lin- cells (n = 6/group, mean±s.e.m., unpaired two-tailed t-test). **e**, Bone marrow engraftment of MHC-II⁺ Lin- cells in different cell types after primary and secondary transplantation; multipotent progenitor (MPP)3/4, MPP2, MPP1, haematopoietic stem cell (HSC), granulocyte-monocyte precursor (GMP) pre-megakaryocyte/erythrocyte (PreMegE), pre- granulocyte monocyte (preGM), pre-colony-forming unit-erythroid cell (CFU-E), CFU-E, LSK-, CD8 T cell (TC), CD4 TC, neutrophil (neutro.), monocyte (mono.), macrophage (macro.) eosinophil (Eos), B cell (BC) and dendritic cell (DC). Illustrations in **b** created using BioRender.com.

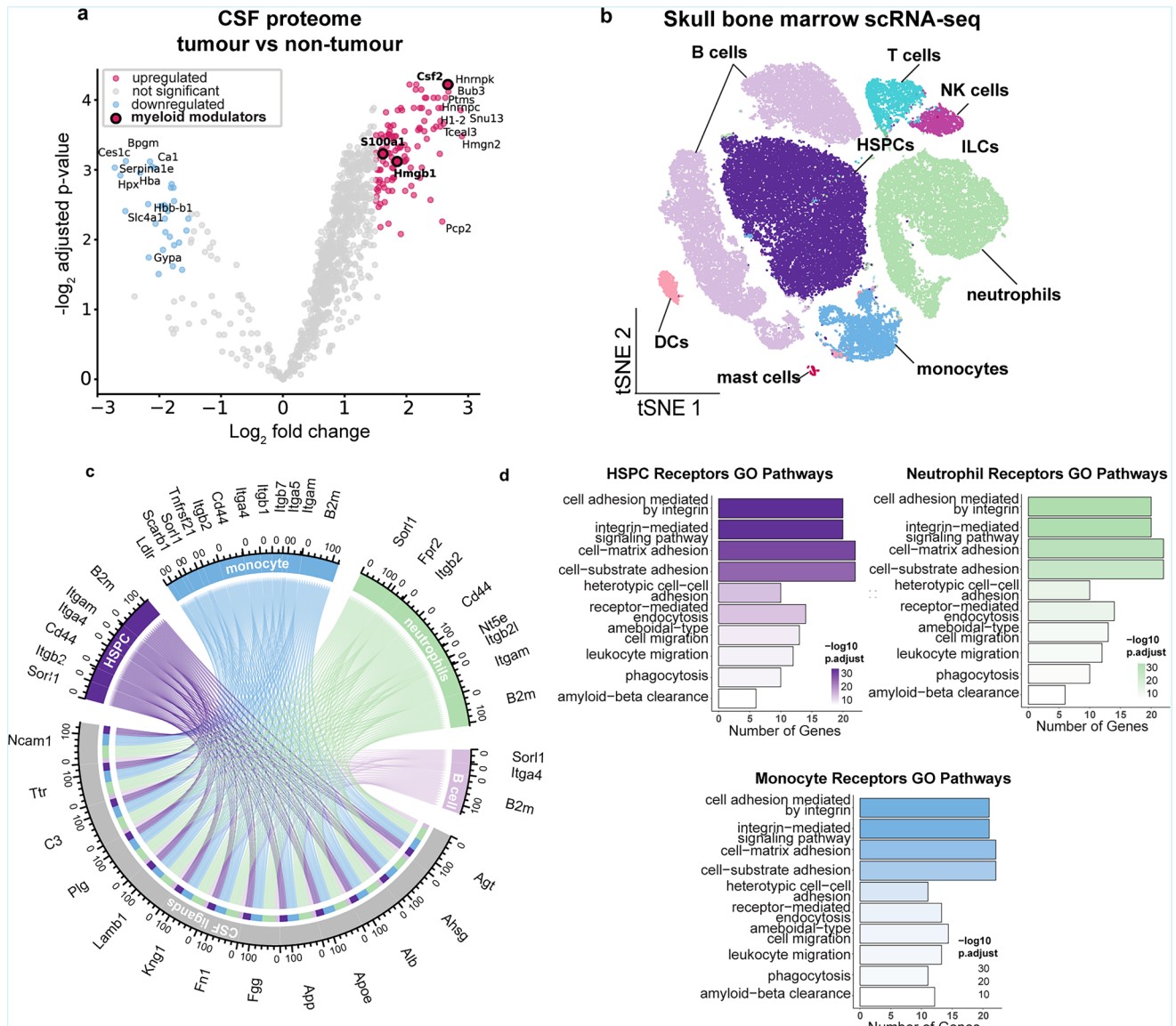

**Extended Data Fig. 2 | Cerebrospinal fluid (CSF) interacts with the skull bone marrow in EP^ZFTA-RELA-bearing mice. a**, Volcano-plot of differential expression analysis of CSF proteins in EP^ZFTA-RELA-bearing relative to control bearing mice (n = 6/group), using a limma-based moderated t-test. **b**, t-distributed stochastic neighbour embedding (t-SNE) visualizations of scRNA-seq from dorsal skull and tibial bone marrow from 2-month-old mice coloured by cell type; B cells, T cells, natural killer (NK) cells, haematopoietic stem progenitor cells (HSPCs), innate

lymphoid cells (ILCs), neutrophils, monocytes, mast cels and dendritic cells (DCs). **c**, Chord plot detailing between CSF ligands identified by tandem mass tag (TMT) Liquid chromatography–mass spectrometry (LC–MS) and receptors on skull bone marrow HSPCs, B cells, monocytes and neutrophils identified by scRNA-seq. **d**, Gene Ontology (GO) pathway analysis on receptor genes with at least one CSF ligand in HSPCs, neutrophils and monocytes.

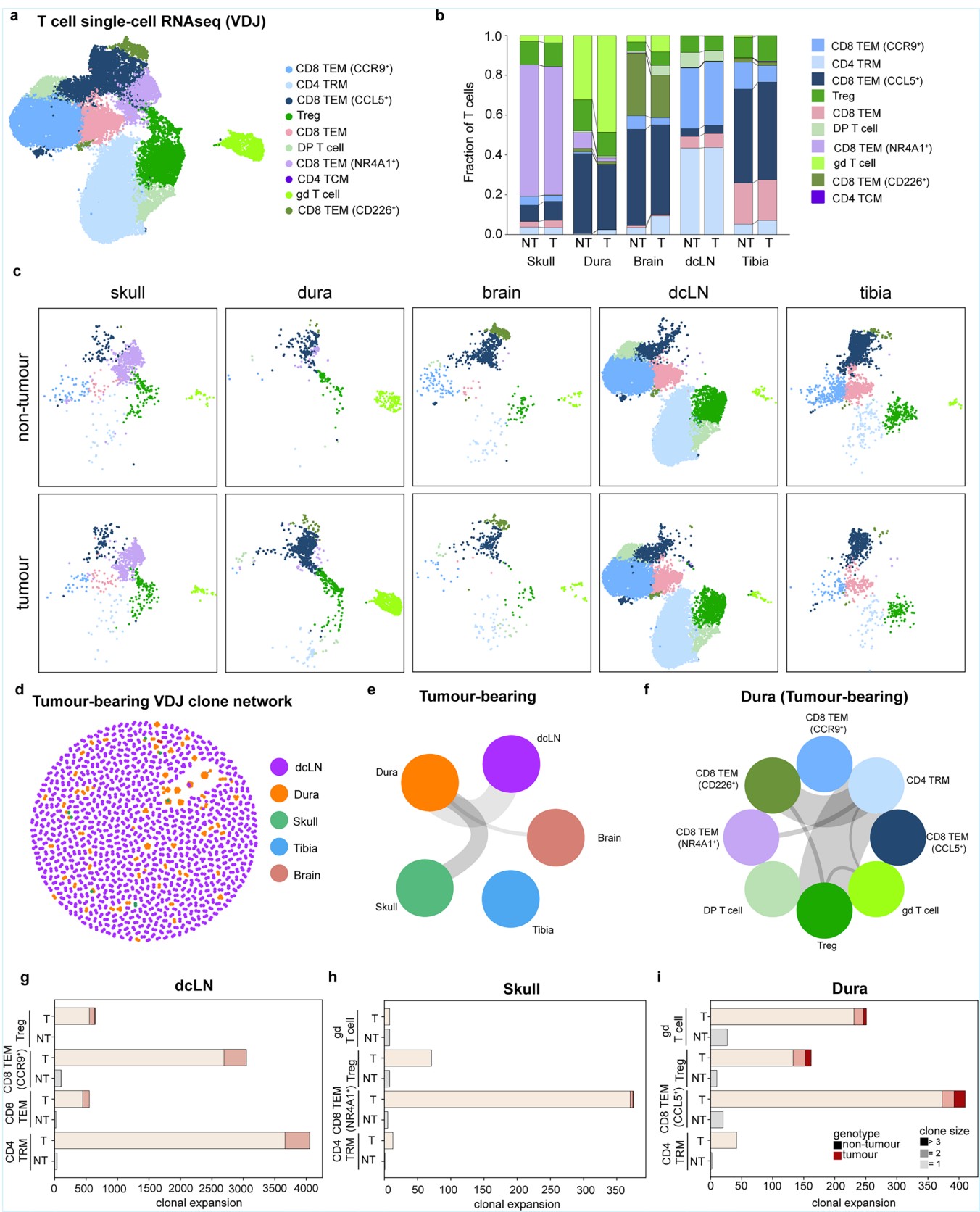

**Extended Data Fig. 3 | See next page for caption.**

**Extended Data Fig. 3 | Clonal expansion of T cells in the dura and skull bone marrow of EP*ZFTA-RELA*-bearing mice. a**, uniform manifold approximation and projection (UMAP) visualisation of single-cell RNAseq of T cell subsets from extravascular CD45$^+$ cells isolated from the skull and tibia bone marrow, dura, deep cervical lymph nodes (dcLN) and brain/tumour from EP*ZFTA-RELA*-bearing and control bearing mice, coloured by cell type; CD8 tissue effector memory (CD8 TEM) CCR9+, CD4 tissue resident memory (CD4 TRM), CD8 TEM CCL5+, regulatory T cell (Treg), CD8 TEM, double-positive (DP) T cell, CD8 TEM (NR4A1+), CD4 tissue central memory (TCM), gamma-delta (gd) T cells, CD8 TEM CD266+). **b**, Stacked barplot of T cell populations across tissues in EP*ZFTA-RELA*-bearing and control bearing mice. **c**, UMAP of single-cell RNAseq of T cell subsets split by tissue and genotype. **d**, variable–diversity–joining rearrangement (VDJ) T cell receptor (TCR) clone network across tissues in EP*ZFTA-RELA*-bearing mice. **e**, circle plot of shared T cell receptor clones between tissues in EP*ZFTA-RELA*-bearing mice. **f**, circle plot of shared T cell receptor clones between cell types in dura of EP*ZFTA-RELA*-bearing mice. **g**–**i**, Bar plot of expanded clones in top 4 expanded T cell populations in each tissue.

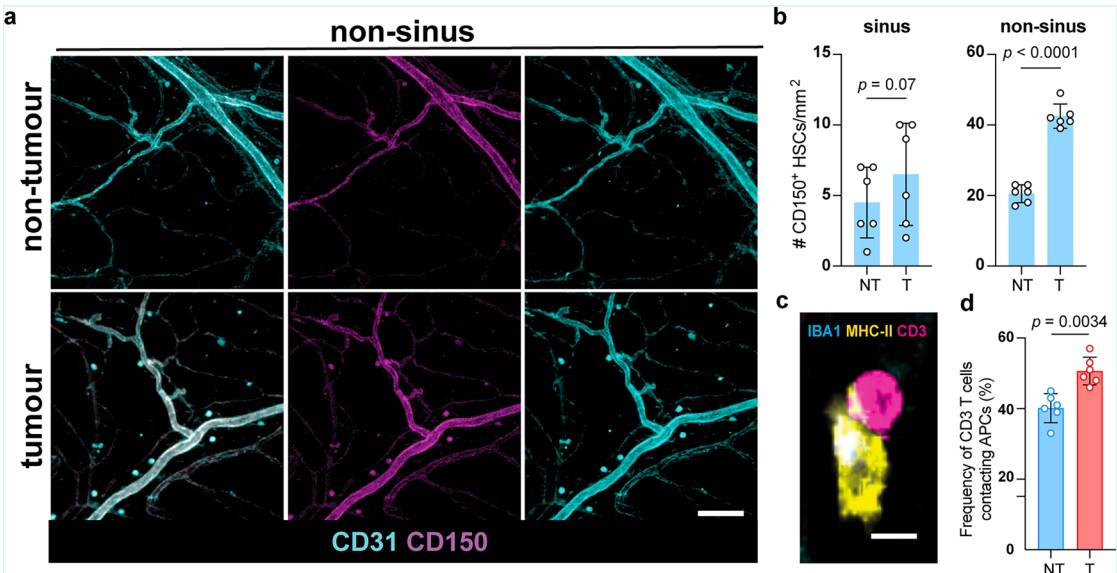

**Extended Data Fig. 4 | Elevated numbers of haematopoietic stem progenitor cells located perivascular non-sinus regions in the dura mater.** **a**, **b**, Immunohistochemistry and quantification of CD150⁺ HSPCs at non-sinus and sinus regions of the dural meninges in EP$^{ZFTA-RELA}$-bearing and control bearing mice (n = 6/group, mean±s.e.m, unpaired two-tailed Student's t-test, scale bar = 100 μm). **c**, **d**, Quantification of CD3⁺ T cells contacting MHC II⁺ cells at dural sinuses, n = 6 mice, n = 382 and n = 460 T cell-MHC II interactions counted total (n = 6/group, mean±s.e.m, unpaired two-tailed Student's t-test, scale bar = 10 μm).

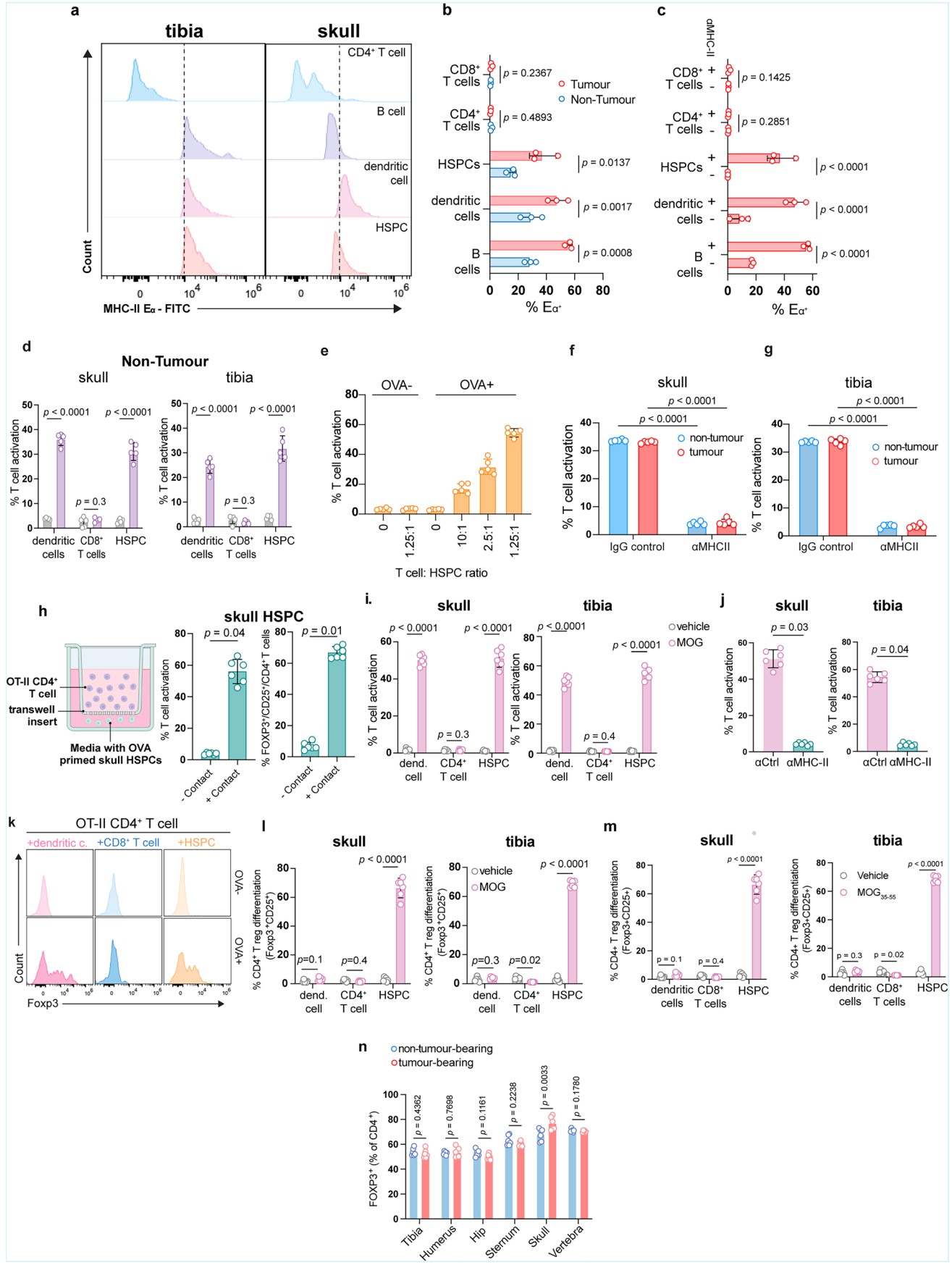

**Extended Data Fig. 5 | See next page for caption.**

**Extended Data Fig. 5 | Skull-derived haematopoietic stem progenitor cells polarise CD4 T cells towards Tregs in an antigen-specific manner.**
**a**, Representative flow cytometry plots of fluorescence-activated cell sorting (FACS) isolated CD4$^+$ T cells, B cells, dendritic cells and haematopoietic stem progenitor cells (HSPCs) from the tibia and skull of tumour and control bearing EP$^{ZFTA-RELA}$ mice pulsed with Eα peptide *ex vivo*. **b**, Quantification of Y-Ae-bound cells from each population as measured by flow cytometry in the presence (**b**) or absence (**c**) of pre-incubation with anti-MHC-II. **d**, T-cell activation (CD44$^+$ cells) after co-culture of skull/tibia-derived DCs, CD8$^+$ T cells, and HSPCs from control bearing mice with OT-II CD4$^+$ T cells (n = 5, 10 pooled mice/replicate; mean±s.e.m.; one-way ANOVA with Šídák's test). **e**, T-cell activation across skull HSPC:OT-II CD4$^+$ ratios. **f**, **g**, T-cell activation in HSPCs ± anti-MHC-II prior to co-incubation with OT-II CD4$^+$ T cells (n = 5; mean±s.e.m.). **h**, Contact-dependent assessment of OT-II CD4 T cell activation and Treg polarization with OVA-primed HSPCs (n = 5/group, mean±s.e.m, unpaired two-tailed Student's t-test). **i**, T-cell activation (CD44$^+$ cells) after co-culture of skull/tibia-derived DCs, CD8$^+$ T cells, and HSPCs from control bearing mice with 2D2 CD4 + T cells (n = 5, 10 pooled mice/replicate; mean±s.e.m.; one-way ANOVA with Šídák's test). **j** T-cell activation in HSPCs ± anti-MHC-II prior to co-incubation with 2D2 CD4$^+$ T cells (n = 5; mean±s.e.m.). **k–m**, Treg polarization (FOXP3$^+$ CD25$^+$ CD4$^+$ T cells) after co-culture of skull/tibia-derived DCs, CD8$^+$ T cells, and HSPCs pulsed *ex vivo* with MOG$_{35-55}$ with 2D2 CD4$^+$ T cells (n = 5, 10 pooled mice/replicate; mean±s.e.m.; one-way ANOVA with Šídák's test). **n**, Quantification of the proportion of FOXP3$^+$ CD4$^+$ T cells across bone marrow compartments in EP$^{ZFTA-RELA}$-bearing and control bearing mice (n = 6 per group, 2 independent experiments), # denotes comparison of non-CNS bone marrow to skull bone marrow in each genotype.

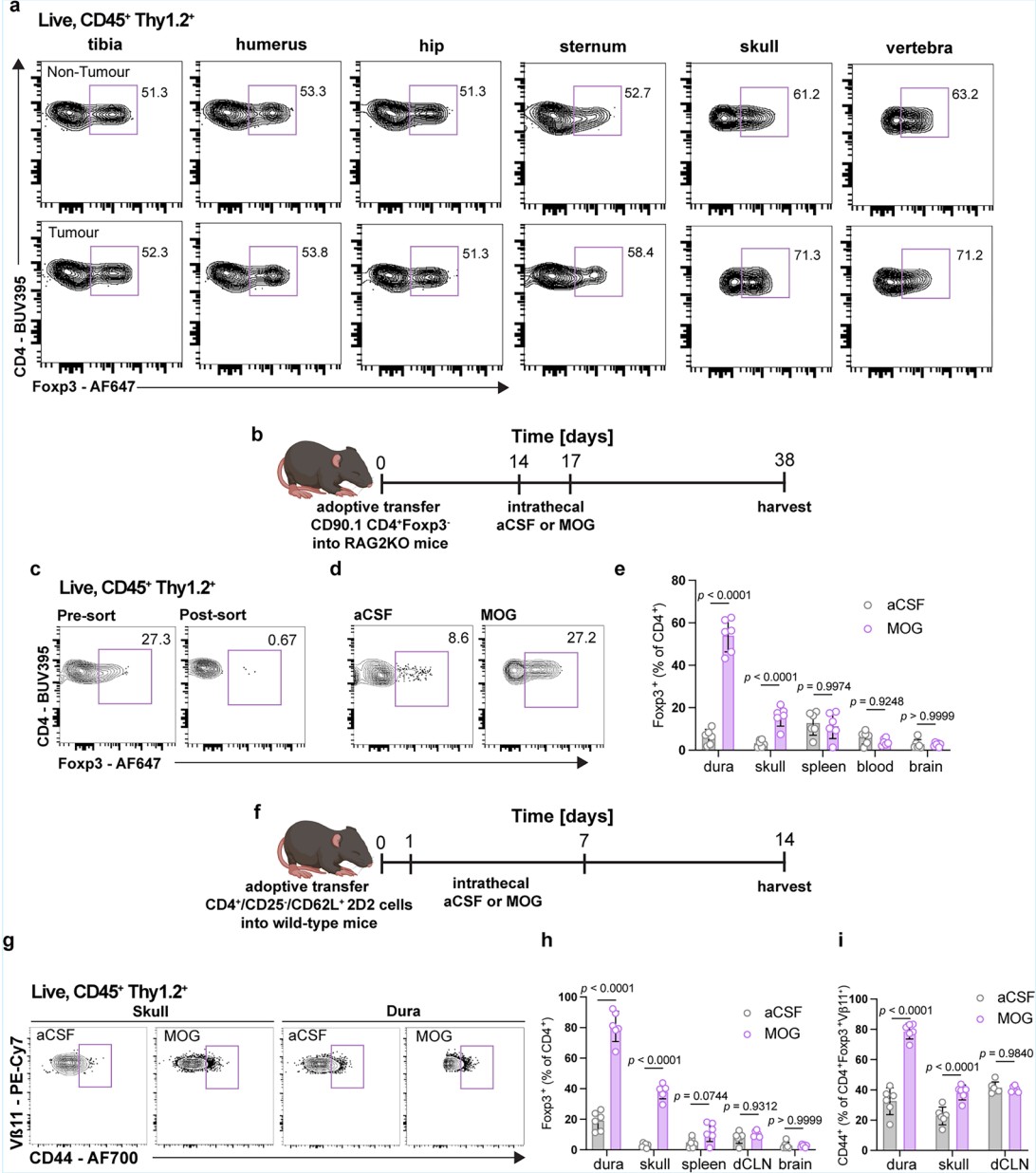

**Extended Data Fig. 6 | Naïve CD4 T cells are polarised to Tregs in the meningeal dura mater and skull bone marrow. a**, Representative flow cytometry analysis of CD4⁺FOXP3⁺ T cells across different bone marrow niches in EP$^{ZFTA-RELA}$-bearing and control bearing mice. **b**, Schematic of the adoptive transfer system from non-Treg CD4⁺ T cells (Thy1.2) cells into RAG2 knockout recipients (KO). **c**, Post-sort validation of isolation of CD4⁺Foxp3⁻ Thy1.2⁺ cells. **d**, Flow cytometry analysis and quantification of CD4⁺FOXP3⁺ T cells in different tissues, n = 6/group, Data are means ± S.E.M, p values represent two-way ANOVA with Sidak's post hoc test. **e**, Quantification of the proportion of Foxp3⁺ CD4 T cells across

bone marrow niches in EP$^{ZFTA-RELA}$-bearing and control bearing mice, combined in. n = 8. Data are means ± S.E.M, p values represent two-way ANOVA with Sidak's post hoc test. **f**, Schematic of adoptive transfer experiments using 2D2 naïve (CD4⁺VB5.1⁺CD25⁻CD62L⁺) T cells to wildtype recipients. **g**, Flow cytometry analysis and quantification of Tregs (CD4⁺Foxp3⁺). **h**, and antigen-specific activated Tregs (Foxp3⁺CD44⁺VB11⁺) (**i**), respectively. n = 6, data are means ± S.E.M, p values represent two-way ANOVA with Sidak's post hoc test. Illustration in **b** created using BioRender.com.

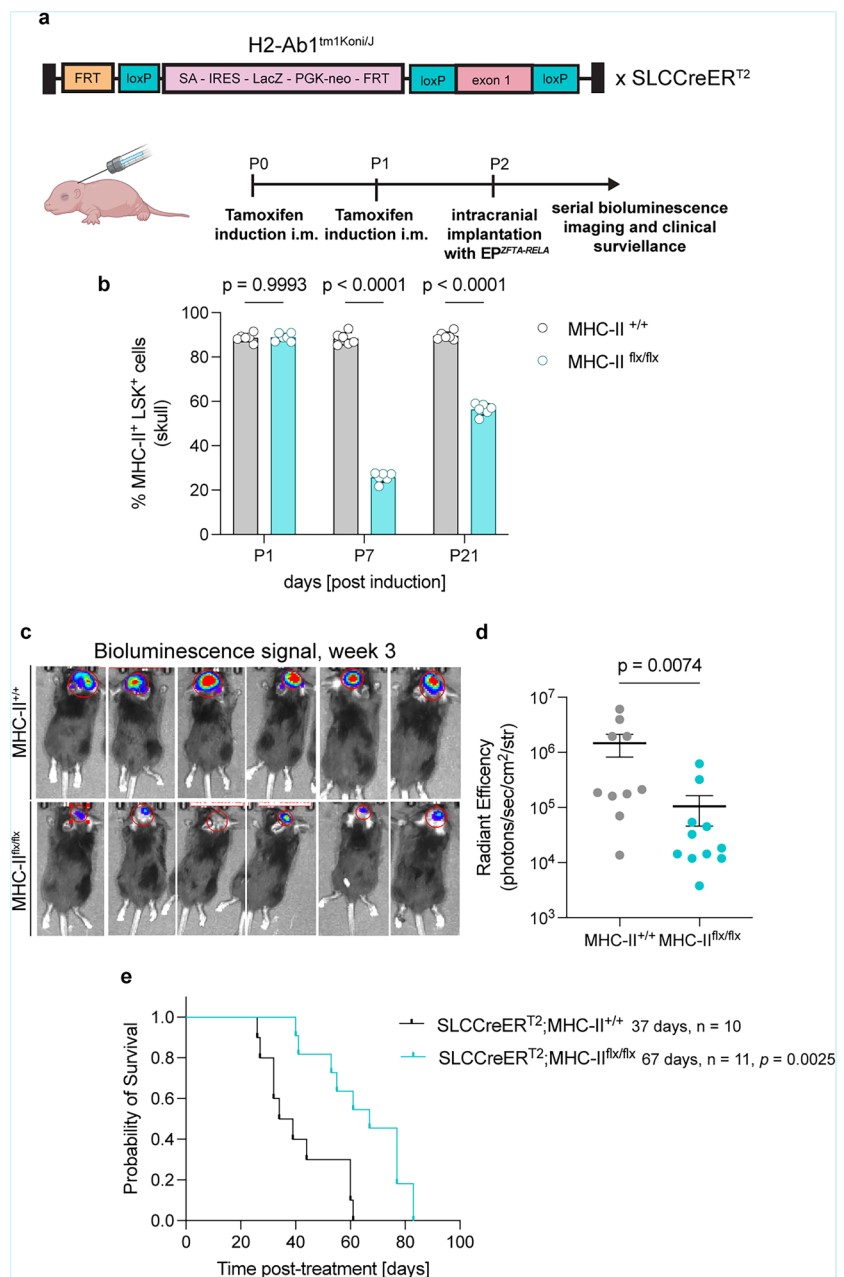

**Extended Data Fig. 7 | Conditional deletion of MHC-II on HSPCs reduces tumour burden in EP[ZFTA-RELA]. a**, Schematic of the H2-Ab1[tm1Koni/J] cassette use to generation the MHC-II[flx/flx] mouse following cross with SLC-CreER[T2] model and induction and implantation timelines. **b**, Quantification of the proportion of MHC-II expressing LSK cells in the skull bone marrow of MHC-II[+/+] and MHC-II[flx/flx] mice up to 21 days (n = 6 per group, 2 independent experiments, data is mean ± SD). **c**, Representative bioluminescent images of 3 week-old mice in each group. **d**, Quantification of bioluminescent signal at 3 weeks after induction and implantation, data is mean±s.e.m, Mann-Whitney, two-tailed t-test. **e**, Kaplan–Meier survival plot of MHC-II[+/+] (n = 10) and MHC-II[flx/flx] (n = 11), Gehan-Breslow Wilcoxon test. Illustration in **a** created using BioRender.com.

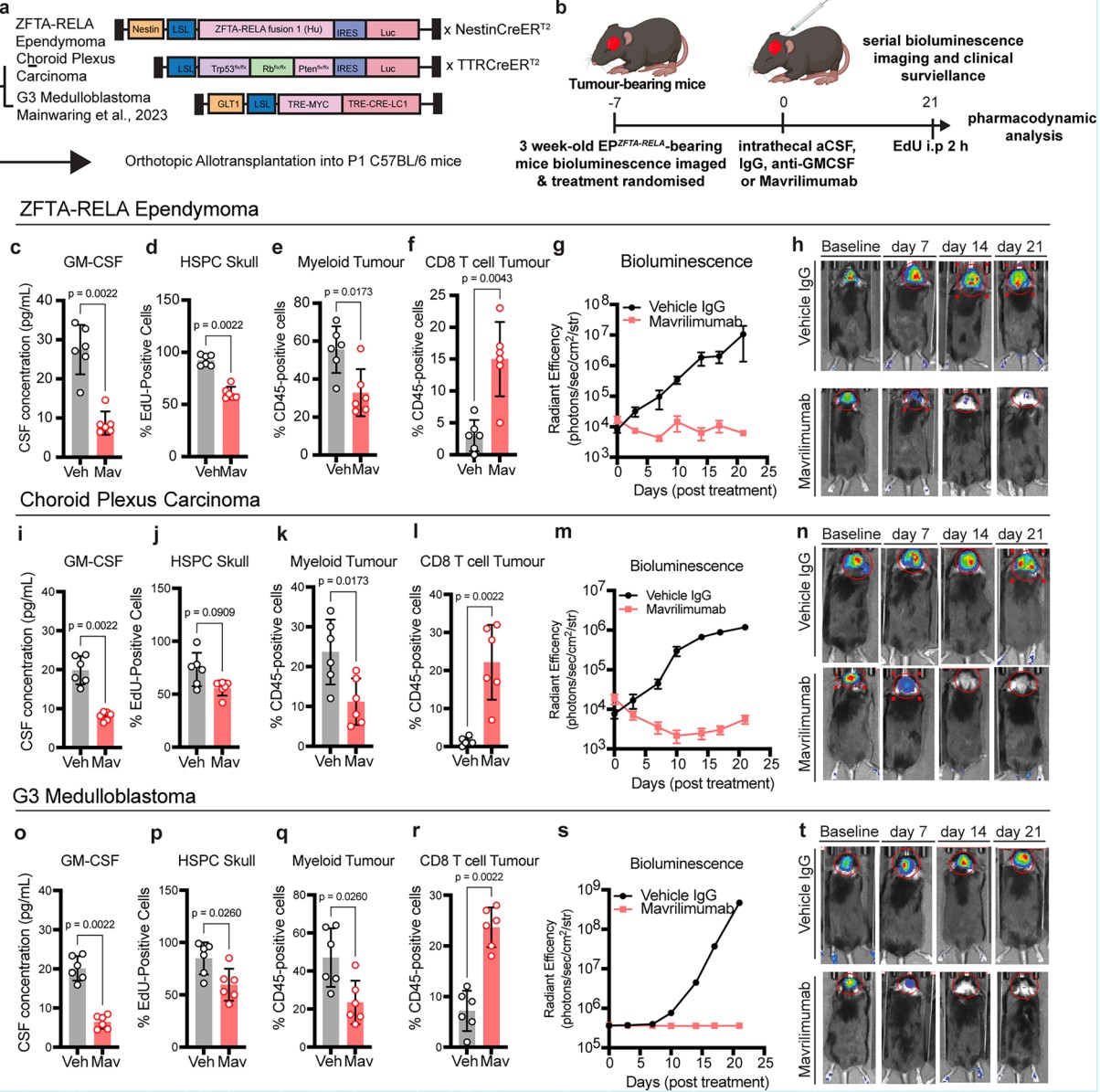

**Extended Data Fig. 8 | Anti-tumour effects of a single intrathecal dose of Mavrilimumab across multiple mouse models of rare childhood brain tumours. a**, schematic of constructs of childhood brain tumour mouse models used in subsequent studies, including ZFTA-RELA ependymoma, choroid plexus carcinoma and group 3 medulloblastoma. **b**, Experimental design for the treatment of 3 week-old EP$^{ZFTA-RELA}$-bearing mice with a single intracisternal magna (ICM) injection of 10 uL of IgG isotype control (Veh, 5 mg/kg, n = 6 per group) or mavrililumab (Mav, 5 mg/kg, n = 6 per group). **c, i, o,**. Enzyme-linked immunosorbent assay (ELISA) quantification of CSF levels of GM-CSF 21 days after intra cisterna magna injection (icm) of Vehicle IgG Isotype control (5 mg/kg) or Mavrililumab (5 mg/Kg) in ZFTA-RELA ependymoma, G3 Medulloblastoma

and Choroid Plexus Carcinoma mouse models. **d, j, p**, Flow cytometry quantification of the perecentage of EdU positive Live CD45$^+$ CD45 i.v- Lineage-Sca1$^+$c-Kit$^+$ HSPCs in the skull bone marrow of mice 21 days post treatment. **e, k, q**, Flow cytometry quantification of the proportion of myeloid (Live CD45$^+$ CD45 i.v- Ly6G- Cd11b$^+$) cells in the tumour parenchyma. **f, l, r**, Flow cytometry quantification of the proportion of CD8 T cells in the tumour parenchyma after treatment for 21 days. **g, m, s**, Tumour progression was then followed by bi-weekly bioluminescence imaging and clinical surveillance. **h, n, t**, Representative bioluminescence signal across one mouse from each group in each tumour model. Data represents, mean ± SD (n = 6/group) unpaired two-tailed Student's t-test). Illustrations in **b** created using BioRender.com.

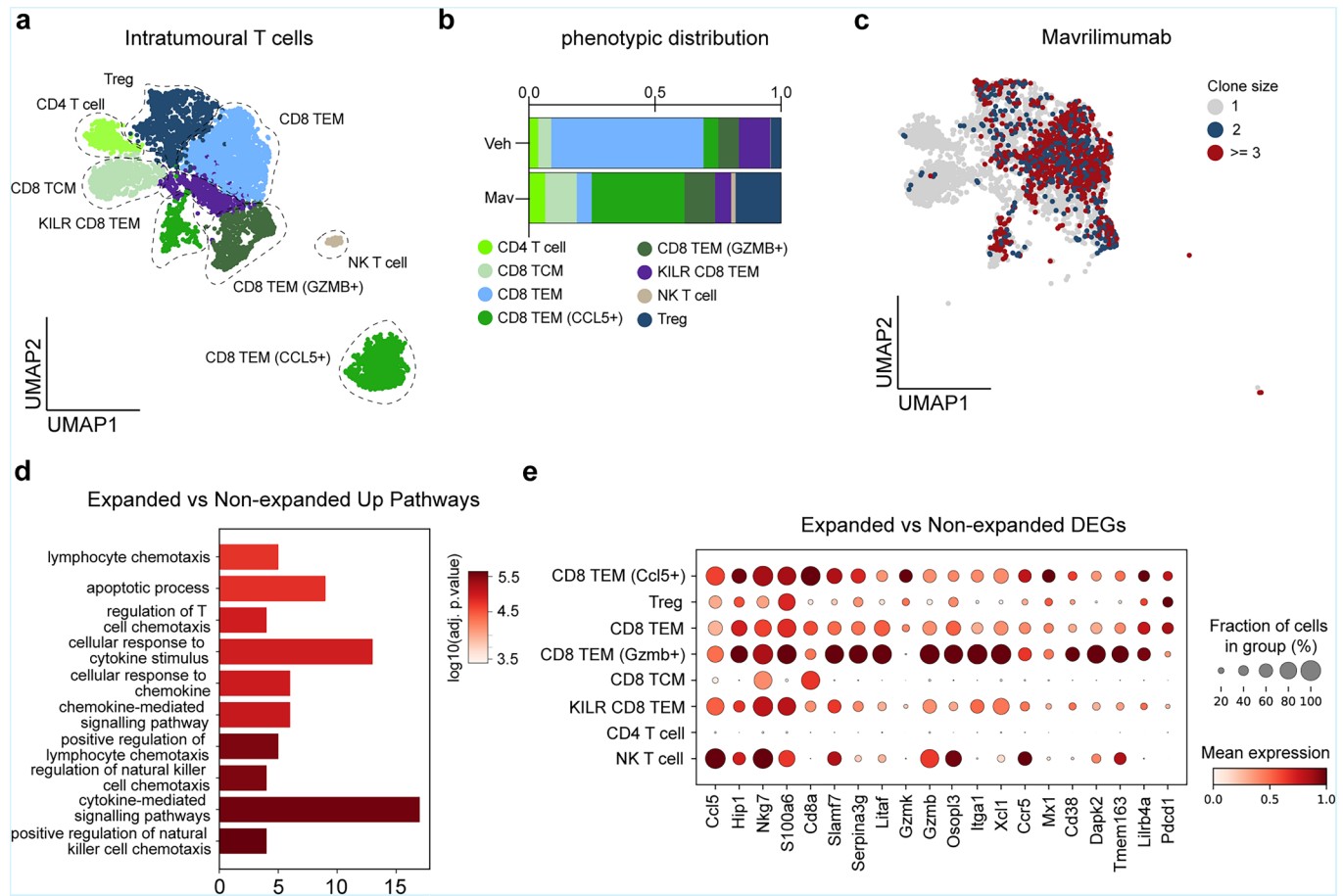

**Extended Data Fig. 9 | Normalizing skull haematopoiesis promotes clonal expansion and tumour infiltration of effector T cells. a**, Uniform-manifold projection of intratumoural T cells in endpoint vehicle and mavrilimumab-treated EP^ZFTA-RELA-bearing mice, coloured by phenotype and quantified in **b**. **c**, UMAP of clone size of intratumoural T cells in mavrilimumab-treated mice.

**d**, Gene Ontology (GO) pathway analysis top differentially upregulated genes in expanded intratumoural T cell clones (>3) relative to non-expanded clones. **e**, dotplot of top 20 differentially upregulated genes across T cell phenotype clusters in mavrilimumab-treated mice, n = 2 pooled experiments per group.

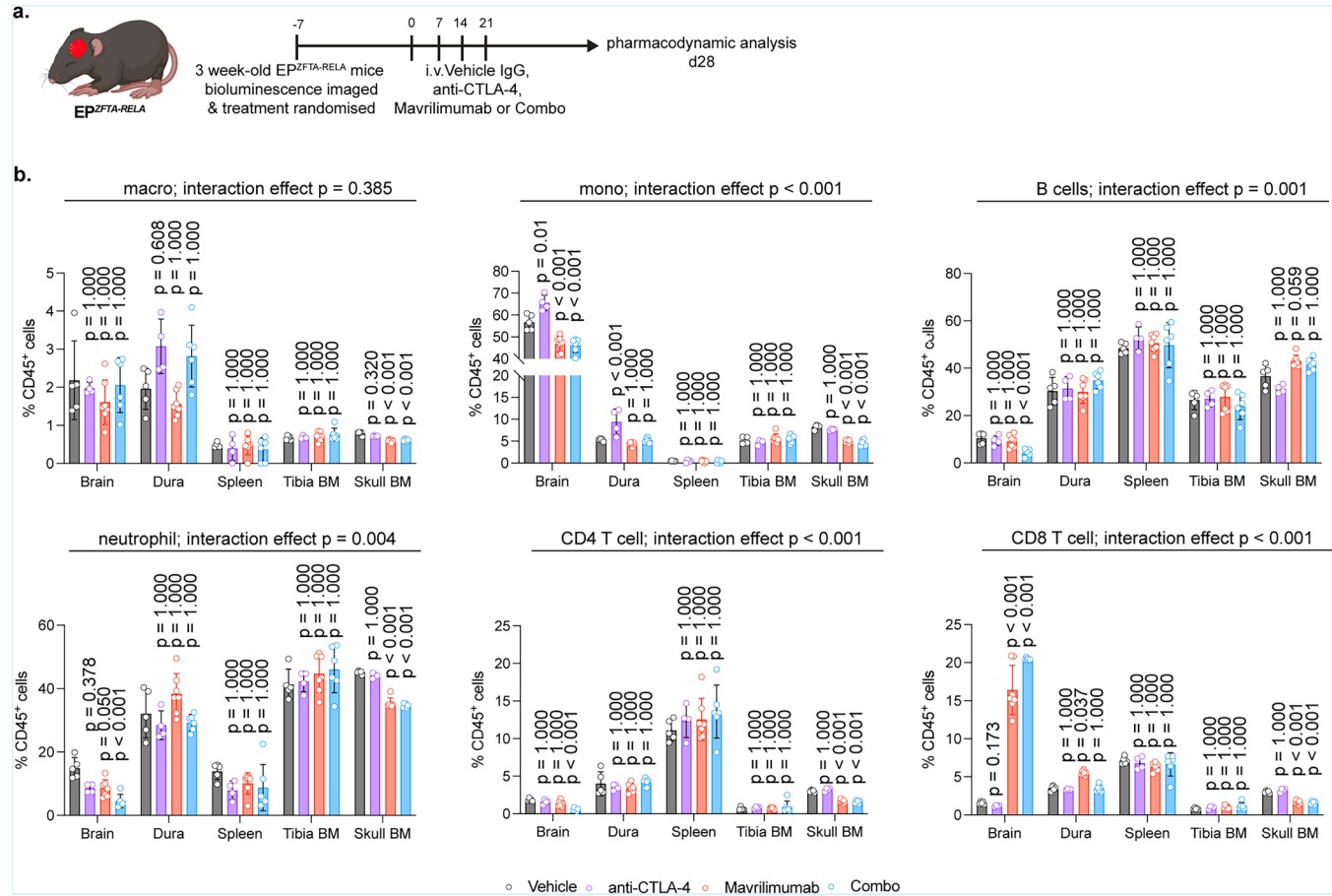

**Extended Data Fig. 10 | Combinatorial treatment of mavrilimumab with immune-checkpoint blockade synergizes for anti-tumour immunity without off-target effects in EP$^{ZFTA\text{-}RELA}$ mice. a**, Experimental design for the treatment of 3 week-old ZFTA-RELA EP$^{ZFTA\text{-}RELA}$-bearing mice with four weekly intravenous injections of Vehicle IgG isotype control (13 mg/kg, n = 5), anti-CTLA-4 (3 mg/kg, n = 4), mavrilimumab (10 mg/kg, n = 7) or combination of anti-CTLA-4 and mavrilimumab (n = 6). **b**, Quantification of the proportion of macrophages,

monocytes, B cells, neutrophils, CD4 T cells and CD8 T cells, within the brain, dura, spleen, tibia and skull bone marrow of treated mice. A linear mixed-effects model was applied to test for interaction across strata, with skull bone marrow as the reference, Wald z-tests on contrasts relative to vehicle with Bonferroni correction for multiple comparisons (two-sided). Illustration in **a** created using BioRender.com.

# Reporting Summary

## Statistics

For all statistical analyses, confirm that the following items are present in the figure legend, table legend, main text, or Methods section.

| n/a | Confirmed | |
|---|---|---|
| ☐ | ☒ | The exact sample size (*n*) for each experimental group/condition, given as a discrete number and unit of measurement |
| ☐ | ☒ | A statement on whether measurements were taken from distinct samples or whether the same sample was measured repeatedly |
| ☐ | ☒ | The statistical test(s) used AND whether they are one- or two-sided *Only common tests should be described solely by name; describe more complex techniques in the Methods section.* |
| ☒ | ☐ | A description of all covariates tested |
| ☐ | ☒ | A description of any assumptions or corrections, such as tests of normality and adjustment for multiple comparisons |
| ☐ | ☒ | A full description of the statistical parameters including central tendency (e.g. means) or other basic estimates (e.g. regression coefficient) AND variation (e.g. standard deviation) or associated estimates of uncertainty (e.g. confidence intervals) |
| ☐ | ☒ | For null hypothesis testing, the test statistic (e.g. *F*, *t*, *r*) with confidence intervals, effect sizes, degrees of freedom and *P* value noted *Give P values as exact values whenever suitable.* |
| ☒ | ☐ | For Bayesian analysis, information on the choice of priors and Markov chain Monte Carlo settings |
| ☒ | ☐ | For hierarchical and complex designs, identification of the appropriate level for tests and full reporting of outcomes |
| ☒ | ☐ | Estimates of effect sizes (e.g. Cohen's *d*, Pearson's *r*), indicating how they were calculated |

*Our web collection on statistics for biologists contains articles on many of the points above.*

## Software and code

Policy information about availability of computer code

| Data collection | SpectroFlo v 2.2.0.3 (Cytek)<br>Leica Stellaris 8 - MP<br>Luminex FLEXMAP 3D |
|---|---|
| Data analysis | Detailed analysis methods can be found in the methods section of the manuscript<br>Packages and software used include: Cell Ranger ARC Suite (v.2.0.0), MACS2(v2.2.7), Seurat v4.3.0, Seuratv5.0.1, SCTransform v2 (v0.4.1), limma (v3.58.1), chromVAR (v1.16.0), ArchR (v1.0.3), scanpy (v1.11.3), Dandelion (v0.5.5)<br>FIJI image processing software (NIH) - v 2.14.0/1.54f<br>Prism v 10.2.0 (GraphPad Software, Inc)<br>FastQC v 0.11.5 - R v 3.5.0<br>RStudio v 1.4.1717<br>Bioconductor DESeq2 v3.5<br>FlowJo software v 10 (BD Biosciences) |

For manuscripts utilizing custom algorithms or software that are central to the research but not yet described in published literature, software must be made available to editors and reviewers. We strongly encourage code deposition in a community repository (e.g. GitHub). See the Nature Portfolio guidelines for submitting code & software for further information.

## Data

Policy information about availability of data

All manuscripts must include a data availability statement. This statement should provide the following information, where applicable:
- Accession codes, unique identifiers, or web links for publicly available datasets
- A description of any restrictions on data availability
- For clinical datasets or third party data, please ensure that the statement adheres to our policy

The accession number for the Fastq files and quantified gene counts for single-cell sequencing reported in this paper is GEO: GSE28237, GSE82459, GSE300889 and GSE300890. All data are available in the main text or the Supplementary Information files. Data were also sourced from the following published accessions: NEMO; dat-oii74w and dat-3ah9h9x, GEO; GSE141460, GSE126025, GSE156053, GSE226961 and GSE231860. The mass spectrometry proteomics data have been deposited to the ProteomeXchange Consortium via the PRIDE partner repository with the dataset identifier PXD058239.

## Research involving human participants, their data, or biological material

Policy information about studies with human participants or human data. See also policy information about sex, gender (identity/presentation), and sexual orientation and race, ethnicity and racism.

| | |
|---|---|
| Reporting on sex and gender | Sex and gender were not used in any scenario as criteria for sample collection. Sex of 2nd trimester de-identified samples were determined based on sex-specific gene expression patterns as determined in previously published datasets. Sex information from all other deidentified samples were provided by the relevant brain and tissue banks. Both male and female samples were treated equally. |
| Reporting on race, ethnicity, or other socially relevant groupings | No race, ethnicity or socially relevant groupings were performed in this study |
| Population characteristics | We collected neurosurgical material from a single 6 mo old M, diagnosed with Atypical choroid plexus papilloma (CNS Who Grade 2), no prior treatments |
| Recruitment | De-identified tissue samples were collected with previous patient consent in strict observance of the legal and institutional ethical regulations. This was performed by the clinic and the inclusion criteria was <18 years of age, primary primary tumour diagnosis. Because we have no demographic information about our sample or the patient population we cannot comment on how any bias may or may not have been present. |
| Ethics oversight | Ethical approval was given through Cambridge central  Research Ethics Committee (23/EE/0241), administrered through Cambridge University Hospitals NHS Foundaation Trust. All subjects provided informed written consent |

Note that full information on the approval of the study protocol must also be provided in the manuscript.

## Field-specific reporting

Please select the one below that is the best fit for your research. If you are not sure, read the appropriate sections before making your selection.

☒ Life sciences      ☐ Behavioural & social sciences      ☐ Ecological, evolutionary & environmental sciences

For a reference copy of the document with all sections, see nature.com/documents/nr-reporting-summary-flat.pdf

## Life sciences study design

All studies must disclose on these points even when the disclosure is negative.

| | |
|---|---|
| Sample size | Sample size: Statistical methods were not used to recalculate or predetermine study sizes but were based on similar experiments previously published (Cugurra et al., Science (2021); Rustenhoven et al., Cell (2021); Mazzitelli et al. (2022)). |
| Data exclusions | No data were excluded for analysis |
| Replication | All experiments were replicated in at least two independent experiments for a total of at least 3 mice per group, and all replication was successful. All representative images are representative of the same experiment performed in at least 3 animals. For in vitro experiments, cells from the same animal were resuspended and plated with treatments and controls in adjacent wells. Single cell RNA sequencing finding have been validated by microscopy and/or flow cytometry as stated in the text. All attempts at replication were successful |
| Randomization | For all experiments, animals from different cages were randomly assigned to different experimental groups. Because all variables were controlled for, no covariates were present. For in vitro studies, cells from the same animal were harvested and resuspended. Treatments and controls were plated adjacent to one another. |
| Blinding | For all experiments, the researchers were blinded, where possible, for at least one of the independent experiments. These experiments included dural wholemount analysis, and anti-GM-CSF treamtments. |

# Reporting for specific materials, systems and methods

We require information from authors about some types of materials, experimental systems and methods used in many studies. Here, indicate whether each material, system or method listed is relevant to your study. If you are not sure if a list item applies to your research, read the appropriate section before selecting a response.

## Materials & experimental systems

| n/a | Involved in the study |
|-----|----------------------|
| ☐ | ☒ Antibodies |
| ☒ | ☐ Eukaryotic cell lines |
| ☒ | ☐ Palaeontology and archaeology |
| ☐ | ☒ Animals and other organisms |
| ☐ | ☒ Clinical data |
| ☒ | ☐ Dual use research of concern |
| ☒ | ☐ Plants |

## Methods

| n/a | Involved in the study |
|-----|----------------------|
| ☒ | ☐ ChIP-seq |
| ☐ | ☒ Flow cytometry |
| ☒ | ☐ MRI-based neuroimaging |

## Antibodies

| Antibodies used | CD11c Armenian Hamster  APC N418 117310 BioLegend 1:100CD11c Armenian Hamster  BV786 HL3 563735 BD Biosciences 1:250 CD161 (NK1.1) Mouse BV480 PK136 108736 BioLegend 1:500 Ly-6C Rat APC-Cy7 HK1.4 128026 BioLegend 1:200 |
|---|---|
| | Ly-6G Rat PE-CF594 1A8 562700 BD Biosciences 1:400 |
| | CD163 Rat BUV661 Mac-2-158 156704 BioLegend 1:200 |
| | CD45R (B220) Rat BV750 RA3-6B2 747469 BD Biosciences 1:500 |
| | CD45R (B220) Rat eFluor 615 RA3-6B2 42-0452-82 Invitrogen 1:200 |
| | MHC Class II (I-Ad) Rat Pe-Cy7 M5/114.14.2 60-5321 Cytek Biosciences 1:1000 |
| | MHC Class II (I-AI/I-E) Rat BV650 M5/114.14.2 107641 BioLegend 1:1000 |
| | CD45 Rat violetFluor450 30-F11 75-0451-U100 Tonbo Biosciences 1:500 |
| | CX3CR1 Rat BB515 Z80-50 567809 BD Biosciences 1:400 |
| | CD11b Rat BUV805 M1/70 566416 BD Biosciences 1:1200 |
| | CD206 Rat BV711 C068C2 141727 BioLegend 1:200 |
| | CD3 Rat Spark NIR 685 17A2 100262 BioLegend |
| | CD172a Rat RY586 P84 753227 BD Biosciences 1:100 |
| | CD8 Rat PE-Cy7 2.43 60-1886-U100 Tonbo Biosciences 1:400 |
| | F4-80 Rat redfluor710 BM8.1 80-4801-U100 Tonbo Biosciences 1:100 |
| | CD127 (IL-7Ra) Rat BV786 SB/199 563748 BD Biosciences 1:100 |
| | CD49b Rat eFluor 506 Dx5 69-5971-82 Invitrogen 1:200 |
| | CD47 Rat BB700 miap301 742181 BD Biosciences 1:100 |
| | CD4 Rat BV421 GK1.5 100443 BioLegend 1:5000 |
| | CD4 Rat PE GK1.5 100408 BioLegend 1:200 |
| | CD86 Rat SuperBright 436 B7-2 62-0862-82 Invitrogen 1:200 |
| | P2RY12 Rat APC-Fire 810 S16007D 848013 BioLegend 1:200 |
| | CD69 Armenian Hamster  PerCP H1.2F3 104520 BD Biosciences 1:100 |
| | CD62L (L-Selectin) Rat APC-Cy7 MEL-14 25-0621-U100 Tonbo Biosciences 1:800 |
| | CD44 Rat AlexaFluor 700 IM7 560567 BD Biosciences 1:200 |
| | TCRγδ Armenian Hamster  BUV737 GL3 748991 BioLegend 1:200 |
| | CD25 Rat BV711 PC61 102049 BioLegend 1:400 |
| | CD366 (TIM3) Rat BV605 RMT3-23 119721 119721 1:100 |
| | CD223 (LAG-3) Rat BV650 C9B7W 740560 BD Biosciences 1:100 |
| | CD19 Rat Super Bright 780 1D3 78-0193-82 Invitrogen 1:400 |
| | CD11a Rat BUV 496 2D7 741056 BD Biosciences 1:200 |
| | CD49d Rat PerCP-Cyanine 5.5 R1-2 65-0492-U100 Tonbo Biosciences 1:400 |
| | MHC Class I (HLA-G) Rat BUV 563 M1/42 749703 BD Biosciences 1:400 |
| | CD152 (CTLA-4) Armenian Hamster  PE UC10-4F10-11 553720 BD Biosciences 1:100 |
| | CD274 Rat BUV615 MIH5 752339 BD Biosciences 1:200 |
| | Lineage Rat Pacific Blue 17A2; RB6-8C5; RA3-6B2; Ter-119; M1/70; 133310 BioLegend 1:500 |
| | CD117 Rat BV421 2B8 105828 BioLegend 1:500 |
| | CD117 Rat BV605 2B8 105847 BioLegend 1:300 |
| | Sca-1 Rat Super Bright 660 D7  64-5981-82 Cytek Biosciences 1:500 |
| | Sca-1 Rat AlexaFluor 647 D7 108118 BioLegend 1:200 |
| | FOXP3 Rat AlexaFluor 647 MF23 560401 BioLegend 1:200 |
| | GATA3 Mouse AlexaFluor 488 16E10A23 653807 BioLegend 1:200 |
| | Ki67 Mouse BV605 B56 567122 BD Biosciences 1:100 |
| | RORγt  Mouse BV650 Q31-378 564722 BD Biosciences 1:200 |
| | T-bet Mouse PerCP-Cy5.5 4B10 644806 BioLegend 1:50 |
| | TCRVb Armenian Hamster  BUV805 H57-597 748405 BD Biosciences 1:200 |
| | TCRvB11 Rat Pe-Cy7 KT-11 125916 BioLegend 1:200 |
| | CD64 Mouse BV421 X54-5/7.1 139309 BioLegend 1:200 |
| | CD115 Rat BUV737 AFS98 750948 BD Biosciences 1:200 |

XCR1 Mouse BV510 ZET 148220 BioLegend 1:200
TCRγδ Human REA633 REA633 130117111 MiltenyiBiotec 1:50
RFP Rabbit FITC - ab34764 Abcam 1:200
CD150 Rat BV650 TC15-12F12.2 115932 BioLegend 1:400
CD41 Rat FITC MWReg30 133904 BioLegend 1:250
CD16/32 Rat PeDazzle S17011E  156616 BioLegend 1:200
CD105 Rat RB780 MJ7/18 755709 BD Biosciences 1:100
CD48 Rat BB700 HM48-1 742119 BD Biosciences 1:200
CD34 Rat eFluor450 Ram34 48-0341-82 Invitrogen 1:200
CD184 Rat BUV805 2B11/CXCR4 741979 BD Biosciences 1:800
CD201 Rat PerCP-eFluor 710 eBio1560 (1560) 46-2012-80 Invitrogen 1:400
CD43 Rat PerCP 1B11 121222 BioLegend 1:200

Human
BD Horizon™ BUV737 Mouse Anti-Human CD16 BUV737 564434 BD Biosciences 1:200
BD OptiBuild™ BUV496 Mouse Anti-Human CD44 BUV496 750519 BD Biosciences 1:500
Alexa Fluor® 700 anti-human CD45 Antibody AF700 304024 Biolegend 1:500
PE/Cyanine7 anti-human CD66b Antibody - G10F5 Pe Cy7 305116 Biolegend 1:500
PerCP anti-human CD14 Antibody PerCP 325632 Biolegend 1:500
Brilliant Violet 510™ anti-human CD49d Antibody BV510 304318 Biolegend 1:500
Brilliant Violet 711™ anti-human HLA-DR Antibody BV711 307644 Biolegend 1:500
PE/Dazzle™ 594 anti-human CD56 (NCAM) Antibody PeDazzle 318348 Biolegend 1:500
Brilliant Violet 605™ anti-human CD25 Antibody BV605 302632 Biolegend 1:200
PE/Cyanine5 anti-human CD127 (IL-7Rα) Antibody PeCyanine 5 351324 Biolegend 1:200
PE/Fire™ 810 anti-human CD274 (B7-H1, PD-L1) Antibody PDL-1 329755 Biolegend 1:500
FITC anti-human CD34 Antibody FITC 343604 Biolegend 1:500
PE anti-human CD90 (Thy1) Antibody PE 328110 Biolegend 1:500
Pacific Blue™ anti-human/mouse CD49f Antibody Pac Blue 313620 Biolegend 1:200
Brilliant Violet 650™ anti-human CD4 Antibody BV650 317436 Biolegend 1:1000
Alexa Fluor® 647 anti-human CD8a Antibody AF647  300918 Biolegend 1:500
Brilliant Violet 421™ anti-human CD11c Antibody BV421 301628 Biolegend 1:1000
PerCP/Cyanine5.5 anti-human CD3 Antibody PerCP/Cy5.5 300328 Biolegend 1:2000
BD Horizon™ BUV661 Rat Anti-CD11b BUV661 612977 BD Biosciences 1:500
BD Horizon™ BUV563 Mouse Anti-Human CD19 BUV563 612916 BD Biosciences 1:500
BD OptiBuild™ BUV395 Mouse Anti-Human CD279 (PD-1) BUV395 745619 BD Biosciences 1:200
APC/Cyanine7 anti-human CD45RA Antibody APC-Cy7 304128 Biolegend 1:500
Spark NIR™ 685 anti-human CD62L Antibody Spark nir 685 304862 Biolegend 1:1000
APC anti-human CD38 Antibody APC 356606 Biolegend 1:2000
CD61 Armenian Hamster  APC 2C9.G2 (HMβ3-1) 104316 BioLegend 1:200

| | |
|---|---|
| Validation | Each antibody was validated for the species (mouse or human) and application (immunohistochemistry, flow cytometry) by the correspondent manufacturer. The usage was described in full detail the methods section of the manuscript. |

## Animals and other research organisms

Policy information about studies involving animals; ARRIVE guidelines recommended for reporting animal research, and Sex and Gender in Research

| | |
|---|---|
| Laboratory animals | Mice were housed in individually ventilated cages with wood chip bedding plus cardboard fun tunnels and chew blocks under a 12 hour light/dark cycle at 21 ± 2°C and 55% ± 10% humidity. Standard diet was provided with ad libitum water. Mice were allowed to acclimate for at least one week in the animal facility prior to the beginning of any experiment. Adult males and females between 4-6 weeks of age were primarily used for our studies unless stated otherwise. Sample sizes were determined on the basis of a power analysis in accordance with previously published experiments. Experimenters, where necessary, were blinded to experimental groups during both scoring and quantification.  The following strains were used: Rosa-Cre-ERT2 (JAX:008463), Nes-Cre- ERT2 (JAX:003771), C57BL/6 (JAX:000664), C57BL/6-Tg(Tcra2D2,Tcrb2D2)1Kuch/J (JAX:006912), B6.Cg-Tg(TcraTcrb)425Cbn/J (JAX:004194), B6.Cg-Rag2tm1.1Cgn/J (JAX:008449), B6.SJL-Ptprca Pepcb/BoyJ (JAX:002014), C57BL/6-Tg(CAG-OVAL)916Jen/J (JAX:005145), B6.Cg-Rag2tm1.1Cgn/J (JAX:008449) |
| Wild animals | No wild animals were used |
| Reporting on sex | Sex was not considered in study design, however experiments were performed on both sexes and we did not observe sex-specific effects. |
| Field-collected samples | This study did not involve field-collected samples |
| Ethics oversight | All animal work was carried out under the Animals (Scientific Procedures) Act 1986 in accordance with the UK Home office license (Project License PP9742216) and approved by the Cancer Research UK Cambridge Institute Animal Welfare and Ethical Review Board. |

Note that full information on the approval of the study protocol must also be provided in the manuscript.

# Clinical data

Policy information about clinical studies

All manuscripts should comply with the ICMJE guidelines for publication of clinical research and a completed CONSORT checklist must be included with all submissions.

| | |
|---|---|
| Clinical trial registration | *Provide the trial registration number from ClinicalTrials.gov or an equivalent agency.* |
| Study protocol | *Note where the full trial protocol can be accessed OR if not available, explain why.* |
| Data collection | *Describe the settings and locales of data collection, noting the time periods of recruitment and data collection.* |
| Outcomes | *Describe how you pre-defined primary and secondary outcome measures and how you assessed these measures.* |

# Plants

| | |
|---|---|
| Seed stocks | *Report on the source of all seed stocks or other plant material used. If applicable, state the seed stock centre and catalogue number. If plant specimens were collected from the field, describe the collection location, date and sampling procedures.* |
| Novel plant genotypes | *Describe the methods by which all novel plant genotypes were produced. This includes those generated by transgenic approaches, gene editing, chemical/radiation-based mutagenesis and hybridization. For transgenic lines, describe the transformation method, the number of independent lines analyzed and the generation upon which experiments were performed. For gene-edited lines, describe the editor used, the endogenous sequence targeted for editing, the targeting guide RNA sequence (if applicable) and how the editor was applied.* |
| Authentication | *Describe any authentication procedures for each seed stock used or novel genotype generated. Describe any experiments used to assess the effect of a mutation and, where applicable, how potential secondary effects (e.g. second site T-DNA insertions, mosiacism, off-target gene editing) were examined.* |

# Flow Cytometry

## Plots

Confirm that:

☒ The axis labels state the marker and fluorochrome used (e.g. CD4-FITC).

☒ The axis scales are clearly visible. Include numbers along axes only for bottom left plot of group (a 'group' is an analysis of identical markers).

☒ All plots are contour plots with outliers or pseudocolor plots.

☒ A numerical value for number of cells or percentage (with statistics) is provided.

## Methodology

| | |
|---|---|
| Sample preparation | Age-matched 6-8 week-old EPZFTA-RELA and NestinCreERT2 mice were intravenously injected with CD45-PE 3 minutes prior to schedule 1. For blood, a single eye was removed using fine curved forceps, rupturing the retro-orbital sinus. Three drops of blood were collected into 1 mL of PBS with 0.025% heparin to prevent coagulation. Samples were kept on ice, for the entirety of collection. Blood was centrifuged at 400 x g for five minutes, and red blood cell (RBC) lysis was performed by resuspension in 1 mL of ACK lysis buffer(Quality Biological) for one minute, then 2 mL of ice-cold PBS was added, samples were centrifuged, and lysed red blood cells were aspirated from the leucocyte-containing pellet. The pellet was resuspended in fluorescence activated cell sorting (FACS) buffer (0.1 M, pH 7.4 PBS with 1% BSA and 1 mM EDTA) until use. Meningeal dura was carefully collected under a dissection microscope. Meninges and calvaria were then digested for 15 minutes at 37°C with constant agitation using 1 mL of pre-warmed digestion buffer (DMEM, with 2% FBS, 1 mg/mL collagenase D (Sigma Aldrich), and 0.5 mg/mL DNase I (Sigma Aldrich)), filtered through a 70 µm cell strainer, and enzymes neutralized with 1 mL of complete medium (DMEM with 10% FBS). An additional 2 mL of FACS buffer was added, samples were centrifuged at 400 x g for five minutes, and samples were resuspended in FACS buffer and kept on ice until use. For peripheral bone marrow, both tibia were flushed with 0.05% BSA PBS with 0.05% EDTA, filtered through 100 µm meshes and washed with 2% fetal bovine serum in RPMI and resuspended on 0.05% BSA PBS solution. The whole intact deep cerbical lymph nodes were mashed through a 70 µm cell strainer, using a sterile syringe plunger, and washed with 5 mL of FACS buffer. Deep cervical lymph node samples were then filtered through 100 µm meshes and washed with 2% fetal bovine serum in RPMI and resuspended on 0.05% BSA PBS solution. Brains were macrodissected based on TdTomato fluorescent signal to harvest the tumour and region-matched brain in non-tumour bearing animals. Tumour/brain samples were mechanically dissociated using sterile surgical scalpels into ~1 mm3 cubes, and dissociated using the mouse tumour dissociation kit (Miltenyi Biotec) using the gentleMACS Octo Dissociator (Miltenyi Biotec). After dissociation, the samples were then filtered through 100µm meshes and washed with 2% fetal bovine serum in RPMI, spun down 420 g for 5 minutes. Samples were resuspended in 40% percoll and centrifuged at 600 x g for 10 minutes. Supernatant was removed and washed with 2% fetal bovine serum in RPMI and resuspended on 0.05% BSA PBS solution. Samples were stained with DAPI (0.2 µg/ml). Samples were centrifuged, resuspended in FACS buffer with anti-CD16/32 (FC block; Biolegend) diluted 1:50 in FACS buffer. Cell surface stains, diluted appropriately in FACS buffer, were then added for 30 minutes on ice. For intracellular staining, suspensions were fixed and permeabilized using the Transcription Factor Fixation/Permeabilization Kit (eBioscience) per the manufacturer's instructions. Antibodies against intracellular proteins, diluted in permeabilization buffer, were added for 30 minutes on ice. |

| | |
|---|---|
| Instrument | Flow cytometry was performed using an Aurora spectral flow cytometer (Cytek Biosciences, CA, USA). |
| Software | Data were collected on SpectroFlo (v2.2.0.3; Cytek) and analyzed with FlowJo (v10; BD Biosciences, NJ, USA). |
| Cell population abundance | For each individual experiment, single-cell suspensions were incubated with viability dyes. Positive populations were gated based on negative control staining. In general, populations are given as a percentage of live, CD45+ cells. |
| Gating strategy | CD45− cells Zombie NIR−, CD45−<br>Myeloid cells Zombie NIR−, CD45+, CD11B+<br>Lymphoid cells Zombie NIR−, CD45+, CD11B−<br>Ly6Chi monocytes: Live/CD45+/CD11b-/Ly6G-/CD19-/TCR-b+/CD4+/GATA3+<br>Neutrophils Zombie NIR−, CD45+, CD11B+, Ly6G+<br>Monocyte-derived macrophages (MDMs) Zombie NIR−, CD45+, CD11B+, Ly6G- CD49Dhigh<br>Microglia (MG) Zombie NIR−, CD45+, CD11B+, CD49Dlow P2RY12+, CX3CR1+<br>Dendritic cells (DCs) Zombie NIR−, CD45+, CD11B+, CD49Dmed, MHC-II+, CD11C+<br>B cells Zombie NIR−, CD45+, CD11B−, CD19+, CD3−<br>− Zombie NIR−, CD45+, CD11B−, CD19−, CD3−<br>CD3+ T cells Zombie NIR−, CD45+, CD11B−, CD19−, CD3+<br>gd T cells Zombie NIR−, CD45+, CD11B−, CD19−, CD3+, gdTCR<br>NK cells Zombie NIR−, CD45+, CD11B−, CD19−, CD3−, CD161+<br>− Zombie NIR−, CD45+, CD11B−, CD19−, CD3+, CD4+, CD8−<br>Double-negative T cells (DNTs) Zombie NIR−, CD45+, CD11B−, CD19−, CD3+, CD4−, CD8−<br>CD8+ T cells Zombie NIR−, CD45+, CD11B−, CD19−, CD3+, CD4−, CD8+<br>CD4+ T cells Zombie NIR−, CD45+, CD11B−, CD19−, CD3+, CD4+, CD8−, CD25−<br>Regulatory T cells (Tregs) Zombie NIR−, CD45+, CD11B−, CD19−, CD3+, CD4+, CD8−, CD127low, CD25+ Foxp3+<br>Haematopoietic stem progenitor cells (HSPCs) Zombie NIR−, CD45+, Lineage-, Sca1+, C-Kit+<br>LS-K cells: Zombie NIR−, CD45+, Lineage-, Sca1-, C-Kit+<br>MPP2 cells: Zombie NIR−, CD45+, Lineage-, Sca1+, C-Kit+, CD150+, CD48+<br>MPP3/4 cells: Zombie NIR−, CD45+, Lineage-, Sca1+, C-Kit+, CD150-, CD48+<br>HSCs: Zombie NIR−, CD45+, Lineage-, Sca1+, C-Kit+, CD150+,CD48-, CD34+<br>GMP: Zombie NIR−, CD45+, Lineage-, Sca1-, C-Kit+, CD16/23+, CD150-<br>CFU-E: Zombie NIR−, CD45+, Lineage-, Sca1-, C-Kit+, CD16/23-, CD150-, CD105+<br>PreCFU-E: Zombie NIR−, CD45+, Lineage-, Sca1-, C-Kit+, CD16/23-, CD150+, CD105+<br>Pre-GM: Zombie NIR−, CD45+, Lineage-, Sca1-, C-Kit+, CD16/23-, CD150-, CD105-<br>PreMegEZombie NIR−, CD45+, Lineage-, Sca1-, C-Kit+, CD16/23-, CD150+ CD105-<br>Th17: Live/CD45+/CD11b-/Ly6G-/CD19-/TCR-b+/CD4+/Ror-gt+/Foxp3-<br>Treg: Live/CD45+/CD11b-/Ly6G-/CD19-/TCR-b+/CD4+/Foxp3+<br>Th1 : Live/CD45+/CD11b-/Ly6G-/CD19-/TCR-b+/CD4+/T-bet+<br>Th2: Live/CD45+/CD11b-/Ly6G-/CD19-/TCR-b+/CD4+/GATA3+ |

☒ Tick this box to confirm that a figure exemplifying the gating strategy is provided in the Supplementary Information.

