## [Peer Review File · Nature Genetics]

Childhood brain tumors instruct cranial haematopoiesis and immunotolerance

Corresponding Author: Professor Richard Gilbertson

This manuscript has been previously reviewed at another journal. This document only contains information relating to versions considered at Nature Genetics.

Version 0:

Decision Letter:

Our ref: NG-A70460-T

12th Nov 2025

Dear Professor Gilbertson,

I am writing on behalf of my colleagues Dr Tiago Faial and Chiara Anania who have been handling your paper.

Thank you for submitting your revised manuscript "Childhood brain tumours instruct cranial haematopoiesis and immunotolerance" (NG-A70460-T). It has now been seen by the original referees and their comments are below. The reviewers find that the paper has improved in revision, and therefore we'll be happy in principle to publish it in Nature Genetics, pending minor revisions to satisfy the referees' final requests and to comply with our editorial and formatting guidelines.

Sincerely,

Safia Danovi
Senior Editor
Nature Genetics
<https://orcid.org/0000-0003-0864-1200>

Reviewer #2 (Remarks to the Author):

We thank the authors for their comprehensive and thoughtful responses to our previous comments provided in the latest round of revisions. The manuscript already contained an extensive body of data in the prior round. We appreciate the authors' efforts to expand their bioinformatic method description, particularly regarding the integration of data sets and VDJ analysis.

Minor points:

- Regarding the integrated human scRNA-seq data set: Fig. 1e and Supp. Fig. 3c-d appear to represent the same UMAPs but with different annotations. We understand that the previous Supplementary Fig. 3b was removed as a duplicate of Fig.

1e. In that case, Supplementary Fig. 3c–d should be oriented identically to Fig. 1e for consistency and ease of interpretation. It is understood, that these data are intended only to demonstrate the presence of HSPCs in human data sets rather than to infer trajectories (as done in mice). Still, the relatively high HSPC levels observed in non-malignant human brain samples contrast with the finding that HSPCs disappear in non-tumor-bearing mice after birth. Are these believed to be circulating cells that were excluded in the mouse flow cytometry setup? This limitation should be acknowledged in the discussion. Additionally, the figure legend should be updated to match the revised panels.

- In response to our previous comment #9, the authors state that the sentence “The combination of mavrimumab and anti-CTLA-4 provide a rationale combination treatment that could be readily translated to the clinic to treat children with a broad array of brain tumours” has been rephrased; however, the wording appears unchanged. As agreed by the authors in the rebuttal letter this statement appears to general give that the combination treatment was only tested in the EPN, but not the other brain tumor models.

Overall, we believe that the authors have assembled a compelling body of evidence that elucidates intricate immune circuits altered by childhood brain tumors within what was formerly thought to be an immune-privileged environment.

Reviewer #3 (Remarks to the Author):

The authors have made substantial improvements to this manuscript and it is an important contribution to the field. I have no further concerns.

Reviewer #4 (Remarks to the Author):

In this manuscript, Cooper et al investigate the regulation of immunosurveillance in the brain. They use a novel genetically engineered mouse model of ZFTA-RELA ependymoma to characterise an immunotolerance circuit in which HSPCs present antigens in the CSF to CD4+ T cells, inducing myeloid bias and CD4+ T cell polarization to Tregs. Treatment with mavrimumab, an antibody directed against GM-CSF disrupted this circuit enabling tumor regression in three tumor models, and the potency of this treatment was enhanced by co-application of anti-CTLA-4. The findings are already prepared for clinical translation in a clinical trial.

Identifying the mechanisms that contribute to immunotolerance in brain tumors are very important in order to enhance treatment opportunities for these cancers with bad prognosis. The here reported findings are highly original and very significant. The methods employed are state-of-the-art and very elaborate, the presentation of the findings is clear and well-contextualized within the current knowledge in the field, the manuscript is overall of very high quality and has much benefitted from the previous review process and the extensive revisions.

The authors have addressed all my concerns appropriately.
